# ADAPTIVE RESIDUAL-UPDATE STEERING FOR LOW-OVERHEAD HALLUCINATION MITIGATION IN LARGE VISION LANGUAGE MODELS

## ABSTRACT

Large Vision-Language Models (LVLMs) often suffer from object hallucination, generating text inconsistent with visual inputs, which can critically undermine their reliability. Existing inference-time interventions to mitigate this issue present a challenging trade-off: while methods that steer internal states or adjust output logits can be effective, they often incur substantial computational overhead, typically requiring extra forward passes. This efficiency bottleneck can limit their practicality for real-world, latency-sensitive deployments. In this work, we aim to address this trade-off with **Residual-Update Directed DEcoding Regulation (RUDDER)**, a low-overhead framework that steers LVLMs towards visually-grounded generation. RUDDER is built on two key innovations: (1) Contextual Activation Residual Direction (CARD) vector, a per-sample visual evidence vector extracted from the residual update of a self-attention layer during a *single, standard forward pass*. (2) A Bayesian-inspired adaptive gate that performs token-wise injection, applying a corrective signal whose strength is conditioned on the model's deviation from the visual context. Extensive experiments on key hallucination benchmarks, including POPE and CHAIR, indicate that RUDDER achieves performance comparable to state-of-the-art methods while introducing negligible computational latency, validating RUDDER as a pragmatic and effective approach for improving LVLMs' reliability without a significant compromise on efficiency. Code is available at `https://anonymous.4open.science/r/RrUuDdDdER-1C13/`.

## 1 INTRODUCTION

While Large Vision-Language Models (LVLMs) have shown remarkable capabilities in multimodal tasks and are increasingly deployed to assist with real-world problems (Alayrac et al., 2022; Liu et al., 2024a), their practical reliability is critically undermined by a persistent challenge: **object hallucination**. As shown in Figure 1, LVLMs frequently generate fluent, convincing text that is factually inconsistent with visual groundings, severely limiting their real-world utility and credibility (Ji et al., 2023). The cause of LVLMs' hallucination lies in the misalignment of information across different modalities: a tendency for powerful pre-trained language models to over-rely on parametric knowledge and language priors at the expense of visual context (Li et al., 2025). To address this without costly retraining, many efforts have focused on inference-time interventions (ITI). However, existing ITI methods present a trade-off between effectiveness and efficiency. These solutions typically fall into two categories: **Non-steering methods** operate on the final output logits. They adjust token probabilities by contrasting different conditions, such as outputs from different model layers (Chuang et al., 2023; Leng et al., 2023). **Steering-based methods** directly modify the model's internal hidden states, allowing them to better align with visual information during generation (Li et al., 2025). While often effective, both approaches share a significant drawback: high computational overhead. They frequently require multiple forward passes through the model, which can double inference latency and make them impractical for real-time applications. This leaves a critical need for a method that is both effective and efficient.

Building on this observation, we argue that a desirable intervention should not force a choice between high performance and practical efficiency. Instead, it should be both **effective and**

Figure 1: **(Left)** An example where the vanilla LLaVA-1.5–7B (Liu et al., 2024a) hallucinates objects. Erroneous text is marked in red, while RUDDER's corrected, factual output is in blue. **(Right)** A comparison showing that unlike existing non-steering and steering-based methods, RUDDER provides adaptive, low-overhead control without requiring extra forward passes.

**lightweight**, operating within a single forward pass, and **context-specific**, capable of adjusting its intensity at each generation step. This requires identifying a reliable signal within the model's internal computation that can correlate with its generation state. Thus, our research is guided by the question: *Can we identify a stable, informative and low-cost signal within the model's standard computational flow to ground generation without introducing extra forward passes?*

We propose **Residual-Update Directed DEcoding Regulation (RUDDER)**, a framework designed to sidestep the efficacy-efficiency trade-off. By leveraging a single-pass intervention that steers generation only when it is consistent with the instance-specific evidence, RUDDER achieves performance comparable to costly, state-of-the-art steering methods while introducing negligible computational latency, effectively making it a low-overhead solution.

Our approach is built on two innovations: (1) the **Contextual Activation Residual Direction (CARD) vector**, a per-sample visual evidence vector extracted from a self-attention layer's residual update during a standard forward pass, and (2) the **Beta Gate**: a Bayesian-inspired adaptive gate that performs token-wise injection of the CARD vector, applying a strong corrective signal only when needed. RUDDER thus offers a pragmatic approach towards visually-grounded generation without compromising on deployment feasibility.

Our main contributions are:

1. We propose the CARD vector, a novel and efficient method for extracting a dynamic, per-sample visual steering vector at a negligible additional cost.

2. We introduce the Beta Gate, an adaptive, token-wise gating mechanism that provides a principled and fine-grained intervention.

3. We demonstrate through extensive experiments across LVLMs with distinct architectures that RUDDER significantly reduces object hallucination to a level comparable with state-of-the-art methods, while introducing negligible computational overhead, thereby offering a superior balance between efficacy and efficiency.

## 2 RELATED WORK

Our research is situated at the intersection of inference-time intervention (ITI) and probabilistic gating.

**Inference-time intervention.** ITI aims to guide a model's generative behavior without modifying its weights. We group existing methods based on where they act on the computation path. *Non-steering methods* operate at the output logits. Many of these methods recalibrate final logits to improve visual grounding, but often at the cost of significant latency due to extra forward passes. For instance, VCD (Leng et al., 2023) uses perturbed images to create a negative context, PAI (Liu et al., 2024b) subtracts unconditional (text-only) logits, and MARINE (Zhao et al., 2025) employs a classifier-free guidance style. Similarly, DoLa (Chuang et al., 2023) contrasts deep vs. shallow logits to suppress generic text. More efficient alternatives such as constrained decoding (Hokamp &

Liu, 2017) or post hoc editing (Manakul et al., 2023) are typically less adaptive. *Steering methods* directly modify the hidden representations to guide the generation trajectory. Most of these methods also incur high computational costs on-the-fly. For example, ASD (Su et al., 2025) steers away from a predefined hallucination direction, and VISTA (Li et al., 2025) injects a signal vector computed from activation differences. VTI (Liu et al., 2025) attempts to mitigate this cost by shifting the computational burden to an offline precomputation step.

**Bayesian and Probabilistic Gating.** Our work is also inspired by Bayesian and probabilistic gating for uncertainty modeling. This includes concepts from Evidential Deep Learning (Sensoy et al., 2018), which frames outputs as parameters of a Dirichlet distribution for uncertainty quantification. Other relevant work explores stochastic gates. For instance, Yamada et al. (2020) use stochastic gates based on a relaxation of the Bernoulli distribution for feature selection. More directly related to our method, Beta-LSTM (Song et al., 2019) replaces standard sigmoid gates with ones derived from a Beta distribution, validating the use of Bayesian principles in gating mechanisms.

# 3 OUR METHOD

To mitigate object hallucination without the high computational costs of existing steering methods, we present Residual-Update Directed DEcoding Regulation (RUDDER). RUDDER is a low-overhead guided decoding framework that adaptively steers LVLMs toward visually-grounded generation by injecting a dynamically derived visual evidence vector into each step of the auto-regressive decoding process. Crucially, it delivers context-specific steering with *no* calibration data and *no* extra forward pass.

This section details the components of our method. We begin with a brief overview of the Transformers residual stream. We then describe two core principles of RUDDER: (1) the zero-cost extraction of the **Contextual Activation Residual Direction (CARD)** vector, and (2) **Beta Gate**, an adaptive injection mechanism guided by a Bayesian-inspired gate.

## 3.1 PRELIMINARIES

The decoder in a Transformer-based LVLM operates on a **residual stream**, where each sublayer's output (e.g., self-attention of the decoder layer) is added back to its input. This output, termed the **residual updates** $\Delta^l$, represents the new information contributed at layer $l$. We leverage these updates during the two-stage auto-regressive generation process: **1. Prefill Stage:** The model processes the prefill span, comprising both image tokens and text prompt tokens, in a single parallel forward pass to populate a Key-Value cache. During this mandatory step, we extract the CARD vector by aggregating the self-attention residual updates across all tokens in the prefill span. **2. Decoding Stage**: The model generates the output sequentially, one token at a time. It's during this phase that we employ Beta Gate for adaptive steering.

## 3.2 CARD VECTOR: A ZERO-COST PER-SAMPLE EVIDENCE DIRECTION

**Motivation.** LVLMs fuse visual and textual information through self-attention. The residual update from the self-attention sublayer, therefore, encodes the net effect of the visual context on the representation of each text token. We hypothesize that by aggregating these updates over the image tokens and text prompt tokens in the prefill span, we can obtain a robust, per-sample vector that captures the direction of visual evidence for the specific input (Liu et al., 2024a). Our empirical analysis supports this: the extracted CARD vector creates a systematic, image-conditioned rotation away from a text-only (language prior) direction, and this rotation aligns coherently with the downstream steering mechanism. This confirms the aggregated updates provide a meaningful directional signal rather than random noise (a detailed visualization and quantification is in Appendix A.4).

To identify the optimal layer for extracting the CARD vector, we analyze **internal dynamics of LLaVA-1.5–7B** (Liu et al., 2024a). We find that intervening in the late decoder layers has the greatest potential to influence the model's final output. Full analysis is provided in Appendix B.1, Figure 6a, 6c.

**Extraction.** In a single standard prefill pass with the image and text prompt, we place a lightweight, read-only hook at the *target* decoder layer $l$ and cache the self-attention output for each token $i$ in

the prefill span $\mathcal{T}_{\text{pre}}$, denoted $\mathbf{A}_i^l$. In a pre-norm decoder, the residual update is simply the attention output,

$$\Delta_i^l = \mathbf{A}_i^l, \tag{1}$$

We then pool these updates and apply $L_2$ normalization to obtain a per-sample direction:

$$\mathbf{v}_{\text{CARD}} = \frac{\text{Pool}\left(\{\Delta_i^l\}_{i \in \mathcal{T}_{\text{pre}}}\right)}{\left\| \text{Pool}\left(\{\Delta_i^l\}_{i \in \mathcal{T}_{\text{pre}}}\right) \right\|_2}, \qquad \text{Pool}(\cdot) \text{ can be mean or } \|\Delta_i^l\|\text{-weighted mean.} \tag{2}$$

This entire process occurs within the single prefill pass and introduces negligible overhead, as no additional forward pass or calibration is required.

### 3.3 BETA GATE: ADAPTIVE INJECTION VIA BAYESIAN-INSPIRED GATING

While other steering methods apply a corrective signal with a fixed strength, this can be suboptimal. As shown in the analysis in Appendix B.1 Figure 6b, the directional coherence of internal update vectors tends to collapse in late decoder layers, which suggests that a fixed, global steering direction could be misaligned at certain steps. A strong correction is only needed when the model's generation deviates from the visual evidence. When the generation is already grounded, a strong intervention may harm output quality.

To address this, we introduce **Beta Gate**, a dynamic, adaptive gating mechanism inspired by Bayesian principles. We frame this problem as determining the "probability of visual groundedness" for each token. This probability is represented as a Bayesian update over a latent gate $g_t \in [0, 1]$, which modulates the strength of the corrective signal on a per-token basis.

**Bayesian view and practical gate.** Let $\mathbf{h}_{l,t}$ be the hidden state for generating the answer token $t$ at our target intervention layer $l$, specifically the output of the LayerNorm operation that precedes the self-attention block. We measure its alignment with the visual context via the cosine similarity $s_t = \cos(\mathbf{h}_{l,t}, \mathbf{v}_{\text{CARD}})$. This score indicates how consistent the current generation trajectory is with the visual evidence. Using a Beta–Binomial intuition, we use $s_t$ to parameterize a Beta distribution, and the gate value $g_t$ is taken as its posterior mean. (The detailed motivation and derivation from a Naïve Bayes perspective are provided in Appendix A.2.). The gate's parameters are calculated as:

$$\alpha_t = \text{softplus}(k\, s_t + c), \qquad \beta_t = \text{softplus}(-k\, s_t + c), \qquad g_t = \frac{\alpha_t}{\alpha_t + \beta_t}, \tag{3}$$

Here, $k$ is a sensitivity hyperparameter that controls the steepness of the gate's response to changes in alignment, and $c$ is a concentration parameter that controls its bias.

To ensure stability, we clamp the gate's output to a predefined range, $g_t \in [g_{\min}, g_{\max}]$. This prevents the gate from completely shutting off ($g_t{=}0$) or saturating at the maximum correction ($g_t{=}1$) too readily, making the intervention more robust.

For generating each token $t$ in the answer, the final steering update $\mathbf{v}_t^{\text{steer}}$ combines the adaptive gate with a global cap $\alpha_{\max}$:

$$\mathbf{v}_t^{\text{steer}} = \underbrace{\left(\alpha_{\max}\, g_t\right)}_{\text{adaptive strength}} \mathbf{v}_{\text{CARD}}, \tag{4}$$

This vector is injected into the residual stream immediately after the Self-Attention (SA) operation. The updated hidden state, $\mathbf{h}_{l,t}^{\text{new}}$, is thus computed as:

$$\mathbf{h}_{l,t}^{\text{new}} = \left(\mathbf{h}_{l,t} + \text{SA}(\mathbf{h}_{l,t})\right) + \mathbf{v}_t^{\text{steer}}. \tag{5}$$

The term $\alpha_{\max} g_t$ represents the **adaptive strength** of the intervention, ensuring a strong corrective signal is applied only when needed; the injection is **restricted to the answer span**.

### 3.4 RUDDER

Our complete method, **Residual-Update Directed DEcoding Regulation (RUDDER)**, integrates the CARD vector and the adaptive Beta Gate to mitigate hallucination by steering LVLMs toward visually grounded outputs. As detailed in Algorithm 1, RUDDER can be seamlessly integrated into the standard auto-regressive decoding loop. By operating within a single inference pass, RUDDER mitigates hallucination with negligible computational overhead, resolving the common trade-off between efficacy and efficiency. The overall workflow of this approach is illustrated in Figure 2.

Figure 2: **The overall workflow of RUDDER.** Our method operates in two stages. **(1) Prefill Stage** (Yellow Arrows): We extract CARD vector $\mathbf{v}_{\text{CARD}}$ by first collecting attention-induced residual updates $\Delta_i^l$ from a target layer $l$ for each token $i$ in the prefill span. These updates are then aggregated using pooling and normalization. The final CARD vectors are cached for each (image, prompt) pair. (2) **Decoding Stage** (Orange Arrows): When generating each answer token $t$, the adaptive Beta Gate computes a steering vector $\mathbf{v}_t^{\text{steer}}$, which is then injected into the residual stream to guide the LVLM towards a more visually-grounded output.

## 4 EXPERIMENTS

In this section, we validate RUDDER, demonstrating its ability to mitigate hallucination effectively with negligible computational overhead. We conduct a series of experiments across diverse LVLM architectures and benchmarks to assess the performance, general capabilities, efficiency, and hyperparameter sensitivity.

### 4.1 EXPERIMENTAL SETUP

**Model Architectures.** We evaluate RUDDER on three representative LVLMs with distinct visual-textual alignment mechanisms: **LLaVA-1.5–7B** (Liu et al., 2024a) and **Idefics2–8b–base** (Laurençon et al., 2024) (which both use a linear projection), and **InstructBLIP** (Dai et al., 2023) (which uses a Q-former (Li et al., 2023a)).

**Decoding Strategies.** We validate RUDDER's versatility across three widely used decoding strategies: **greedy decoding**, **beam search** (beam size of 5), and **nucleus sampling** (top-p=0.9; temperature fixed at 1.0 for all scenarios).

**Baselines.** We compare RUDDER with a set of state-of-the-art inference-time intervention methods to demonstrate its superior trade-off between efficacy and efficiency. Baselines include logit-based strategies like **DoLa** (Chuang et al., 2023), **VCD** (Leng et al., 2023), and **PAI** (Liu et al., 2024b); and steering-based interventions like **VISTA** (Li et al., 2025), representing the dominant paradigms in the field. All baseline results were reproduced under identical evaluation settings for a fair comparison, using the authors' publicly available code whenever possible.

**Evaluation Benchmarks.** To rigorously evaluate RUDDER, we use a combination of specialized hallucination benchmarks and a comprehensive benchmark for general multimodal capabilities.

- **Hallucination Benchmarks.** We directly measure object hallucination using two standard benchmarks: **(1) CHAIR** (Rohrbach et al., 2019): The Caption Hallucination Assessment with Image Relevance benchmark evaluates hallucination in open-ended image captioning tasks. We report two metrics: $\text{CHAIR}_{\text{S}} = \frac{|\{\text{captions with} \geq 1 \text{ hallucinated object}\}|}{|\{\text{captions}\}|}$, which measures the rate of hallucination at the sentence level, and $\text{CHAIR}_{\text{I}} = \frac{|\{\text{hallucinated objects}\}|}{|\{\text{mentioned objects}\}|}$ which measures the rate of hallucination at the object level. For both metrics, lower scores indicate better performance. Following the established protocol, we randomly select 500 samples from the MSCOCO 2014 (Lin et al., 2015) validation set, and evaluate them using

the prompt "`Please help me describe the image in detail`" with a maximum generation length of 512 tokens. **(2) POPE** (Li et al., 2023b): The Polling-based Object Probing Evaluation examines object hallucination through targeted yes/no questions, such as "`Is there a <object> in the image?`". Performance is measured by accuracy and F1 score across its random, popular, and adversarial splits in MSCOCO 2014 subset.

- **General Capabilities Benchmark.** To confirm that our hallucination mitigation does not harm the model's overall abilities, we use **MME** (Fu et al., 2024), a challenging benchmark that assesses a model's performance on a wide range of tasks, including color perception, counting, and positioning, to provide a holistic view of its multimodal capabilities.

**Implementation Details.** We optimize hyperparameters on a holdout validation set of 100 MSCOCO 2014 images to balance generation quality and hallucination reduction. The model-specific configurations are as follows: For LLaVA-1.5, we set the injection layer $L = 30$, with Beta-gate parameters $\alpha_{\max} = 20$, $k = 5.0$. For Idefics2, we use $L = 28$, $\alpha_{\max} = 8.0$, and $k = 5.0$. Since InstructBLIP's Q-former architecture is less effective with mid-to-late layer injections, we set its injection layer to $L = 1$, with $\alpha_{\max} = 6.5$, and $k = 8.0$. Across all models, the gate's concentration parameter $c$ is fixed at 1, and the output was clamped to the range $[0, 1]$. These settings define our main adaptive method, **RUDDER-Beta**, while our fixed-strength ablation, **RUDDER-Add**, uses a constant injection strength equal to each model's respective $\alpha_{\max}$ without the adaptive gate.

### 4.2 RESULTS ON HALLUCINATION BENCHMARKS

### 4.2.1 CHAIR: OPEN-ENDED CAPTIONING

On the CHAIR benchmark, which evaluates hallucination in open-ended captioning, RUDDER demonstrates a strong ability to reduce factual errors while preserving caption quality.

A key challenge in hallucination mitigation is the trade-off with recall: aggressive steering can artificially lower hallucination scores by producing overly simplistic captions. To ensure a fair and practical evaluation, we constrain our analysis to configurations that maintain at least 95% of the vanilla model's recall (i.e., $\text{Recall\_\{evaluated methods\}} \geq 0.95 \times \text{Recall\_\{vanilla model\}}$).

Under this constraint, **RUDDER-Beta** consistently outperforms the vanilla baseline across all tested LVLMs and decoding strategies, as shown in Table 1. It achieves average relative reductions of $33.2\%$ in sentence-leve (CHAIR$_\text{S}$) and $28.6\%$ in object-level (CHAIR$_\text{I}$) hallucination.

Compared to strong baselines like VCD and DoLa, our method is consistently superior on both metrics. Furthermore, RUDDER-Beta performs on par with the state-of-the-art VISTA and, on average, yields a greater reduction in object-level hallucinations (CHAIR$_\text{I}$).

RUDDER-Beta's ability to reduce CHAIR$_\text{I}$ more effectively than CHAIR$_\text{S}$ highlights its precision. We attribute this to the token-wise gating mechanism, which selectively amplifies corrections on visually incongruent or content-noun tokens while leaving already grounded tokens largely unperturbed. This allows RUDDER to preferentially suppress object-level hallucinations without degrading overall caption quality and recall.

### 4.2.2 POPE: VISUAL QUESTION ANSWERING

Moving from open-ended captioning to a more constrained task, we next evaluate RUDDER on the POPE benchmark for object probing. This benchmark tests the model's factuality through targeted yes/no questions, offering a different perspective on hallucination. In this setting, RUDDER again demonstrates competitive performance. As shown in Table 2, RUDDER consistently outperforms the vanilla baselines and most competing methods across all tested models. Concretely, RUDDER-Beta improves accuracy by $1.0/0.7/0.5$ absolute points (pp) and F1 by $1.6/1.3/0.14$ pp on LLaVA-1.5, Idefics2, and InstructBLIP, respectively.

Notably, RUDDER-Beta achieves the highest F1-score and accuracy on both LLaVA-1.5 and Idefics2, surpassing strong steering-based methods like VISTA. While its performance on Instruct-BLIP is slightly surpassed by VISTA when employing greedy decoding and nucleus sampling, RUD-

Table 1: **Hallucination evaluation on the CHAIR benchmark.** We compare RUDDER against state-of-the-art training-free methods, with a maximum generation length of 512 tokens. For each metric, the best-performing method is **bolded** and the second-best is underlined.

| Decoding | Method | LLAVA-1.5 (Liu et al., 2024a) | | Idefics2 (Laurençon et al., 2024) | | InstructBLIP Dai et al. (2023) | |
|---|---|---|---|---|---|---|---|
| | | CHAIR$_S$ ↓ | CHAIR$_I$ ↓ | CHAIR$_S$ ↓ | CHAIR$_I$ ↓ | CHAIR$_S$ ↓ | CHAIR$_I$ ↓ |
| Greedy | Vanilla | 48.6 | 13.6 | 46.6 | 14.9 | 39.2 | 12.8 |
| | DoLa (Chuang et al., 2023) | 47.6 | 13.4 | - | - | - | - |
| | VCD (Leng et al., 2023) | 49.8 | 14.5 | - | - | 46.4 | 15.3 |
| | VISTA (Li et al., 2025) | **38.6** | **11.4** | 33.5 | 11.6 | 27.7 | 9.7 |
| | **RUDDER-Beta** (Ours) | 39.5 | **10.5** | **28.4** | **10.9** | **27.1** | **8.5** |
| | **RUDDER-Add** (Ours) | 42.1 | 11.8 | 30.1 | 11.8 | 28.3 | 10.4 |
| Beam Search | Vanilla | 52.8 | 15.6 | 48.6 | 14.5 | 38.2 | 12.7 |
| | VCD (Leng et al., 2023) | 52.4 | 15.5 | - | - | 47.4 | 16.3 |
| | VISTA (Li et al., 2025) | 33.9 | 10.5 | 32.2 | 11.8 | 27.1 | 9.6 |
| | **RUDDER-Beta** (Ours) | **33.1** | **9.3** | **29.2** | **10.1** | **26.2** | **9.5** |
| | **RUDDER-Add** (Ours) | 35.2 | 10.6 | 31.4 | 10.9 | 27.4 | 11.1 |
| Nucleus Sampling | Vanilla | 55.6 | 16.0 | 53.8 | 16.7 | 46.0 | 16.2 |
| | DoLa (Chuang et al., 2023) | 49.3 | 14.8 | - | - | - | - |
| | VCD (Leng et al., 2023) | 57.5 | 17.2 | - | - | 53.3 | 19.8 |
| | VISTA (Li et al., 2025) | 39.2 | 11.9 | 35.5 | 11.8 | 29.0 | 11.3 |
| | **RUDDER-Beta** (Ours) | 39.9 | **11.0** | **34.1** | **11.3** | **28.9** | 13.7 |
| | **RUDDER-Add** (Ours) | 41.6 | 12.1 | 36.5 | 12.9 | 30.1 | 14.4 |

DER remains highly competitive, highlighting its effectiveness as a versatile solution for reducing object hallucination.

### 4.2.3 ANALYSIS OF ADAPTIVE VS. FIXED-STRENGTH STEERING

A key design choice in RUDDER is whether to use the adaptive gate (**RUDDER-Beta**) or a fixed-strength injection (**RUDDER-Add**). Our experiments show a clear trade-off between these variants, guiding the choice based on the task and model architecture.

For **complex, open-ended generation (CHAIR)**, RUDDER-Beta is consistently superior. Its token-wise precision is crucial for suppressing specific hallucinations in long-form text without harming overall recall. In the **simpler, binary-choice POPE task**, the distinction is more nuanced. While RUDDER-Beta remains the top performer on **LLaVA-1.5** and **Idefics2**, RUDDER-Add is competitive and even surpasses RUDDER-Beta on **InstructBLIP**. We hypothesize this is partly because InstructBLIP's Q-Former provides a highly-condensed visual representation that responds well to a uniform steering signal in a simple setting. For single-token "yes/no" answers, the aggressive push from fixed-strength steering can be sufficient and sometimes more beneficial for certain model architectures.

In summary, RUDDER-Beta is recommended for robust and precise control in complex tasks, while the simpler RUDDER-Add is a powerful option for constrained tasks and certain model architectures.

### 4.3 RESULTS ON COMPREHENSIVE BENCHMARKS

To ensure that hallucination mitigation does not compromise general multimodal capabilities, we evaluate RUDDER on **MME benchmark**. The results show that RUDDER successfully reduces hallucinations without sacrificing the overall abilities of the tested LVLMs. As demonstrated in Table 3, both RUDDER-Beta and RUDDER-Add achieve higher MME scores than the vanilla models for Idefics2 and InstructBlip. On LLaVA-1.5, RUDDER's scores are slightly lower than the vanilla model, but the difference is still acceptable.

### 4.4 EFFICIENCY TESTS

A critical advantage of RUDDER is its low computational overhead, making it practical for real-world deployment. Unlike many state-of-the-art intervention methods that require extra forward passes and significantly increase latency, RUDDER is designed to operate within a single generative pass. We measure the practical latency and throughput of RUDDER against vanilla models and other methods, with results presented in Table 4. All experiments are conducted on a single Nvidia A100 GPU with 80 GB VRAM and a batch size fixed at 1. RUDDER-Beta maintains an

Table 2: **Performance on the POPE benchmark across three LVLMs.** The reported values are the mean accuracy and F1 score, aggregated over the random, popular, and adversarial object splits. The best scores are **bolded**, and the second best scores are underlined.

| Decoding | Method | LLAVA-1.5 (Liu et al., 2024a) | | Idefics2 (Laurençon et al., 2024) | | InstructBLIP Dai et al. (2023) | |
|---|---|---|---|---|---|---|---|
| | | Avg. Accuracy ↑ | Avg. F1 ↑ | Avg. Accuracy ↑ | Avg. F1 ↑ | Avg. Accuracy ↑ | Avg. F1 ↑ |
| Greedy | Vanilla | 85.34 | 84.91 | 78.40 | 74.86 | 85.74 | 84.75 |
| | DoLa (Chuang et al., 2023) | 85.51 | 84.96 | - | - | - | - |
| | VCD (Leng et al., 2023) | 85.46 | 84.87 | - | - | 85.79 | 84.89 |
| | PAI (Liu et al., 2024b) | 85.98 | 85.31 | - | - | - | - |
| | VISTA (Li et al., 2025) | 86.21 | 85.42 | 78.28 | 74.66 | **86.25** | **85.06** |
| | **RUDDER-Beta** (Ours) | **86.53** | **86.03** | **78.74** | **76.52** | 86.02 | 84.93 |
| | **RUDDER-Add** (Ours) | 85.92 | 84.98 | 78.43 | 75.91 | 86.05 | 85.05 |
| Beam Search | Vanilla | 85.46 | 84.98 | 78.67 | 77.55 | 84.73 | 84.37 |
| | VCD (Leng et al., 2023) | 85.60 | 85.06 | - | - | 84.95 | 84.59 |
| | PAI (Liu et al., 2024b) | 85.58 | 85.01 | - | - | - | - |
| | VISTA (Li et al., 2025) | 86.10 | 85.35 | 78.40 | 77.31 | 85.64 | 84.61 |
| | **RUDDER-Beta** (Ours) | **86.51** | **86.19** | 79.33 | 77.96 | 85.54 | 84.40 |
| | **RUDDER-Add** (Ours) | 85.98 | 85.02 | 78.91 | 77.60 | **85.71** | **84.75** |
| Nucleus Sampling | Vanilla | 83.00 | 81.08 | 74.84 | 67.78 | 85.50 | 84.52 |
| | DoLa (Chuang et al., 2023) | 82.94 | 81.12 | - | - | - | - |
| | VCD (Leng et al., 2023) | 82.82 | 81.90 | - | - | 85.61 | 84.65 |
| | PAI (Liu et al., 2024b) | 83.17 | 82.14 | - | - | - | - |
| | VISTA (Li et al., 2025) | 83.58 | 82.21 | 74.66 | 67.70 | **86.12** | **85.26** |
| | **RUDDER-Beta** (Ours) | **84.02** | **83.57** | 75.89 | 69.69 | 85.79 | 84.74 |
| | **RUDDER-Add** (Ours) | 83.20 | 82.38 | 74.95 | 67.84 | 85.95 | 84.95 |

Table 3: **Overall performance scores on the MME full evaluation set.** Higher scores indicate better general capability across perception, reasoning, and knowledge-based tasks.

| Decoding | Method | LLAVA-1.5 (Liu et al., 2024a) | Idefics2 (Laurençon et al., 2024) | InstructBLIP (Dai et al., 2023) |
|---|---|---|---|---|
| Greedy | Vanilla | 1745.87 | 1518.84 | 1566.77 |
| | **RUDDER-Beta** | 1724.17 | 1540.56 | 1592.07 |
| | **RUDDER-Add** | 1715.45 | 1526.03 | 1585.28 |
| Beam Search | Vanilla | 1760.20 | 1450.59 | 1539.16 |
| | **RUDDER-Beta** | 1746.66 | 1484.21 | 1565.77 |
| | **RUDDER-Add** | 1738.13 | 1475.80 | 1560.64 |
| Nucleus Sampling | Vanilla | 1752.65 | 1362.45 | 1538.18 |
| | **RUDDER-Beta** | 1721.94 | 1374.77 | 1556.43 |
| | **RUDDER-Add** | 1713.74 | 1364.16 | 1546.01 |

average throughput of **96.0%** compared to vanilla LVLMs. RUDDER-Add is even more efficient as it bypasses the Beta Gate calculation. In contrast, competing methods that require extra forward passes see a significant drop in efficiency. On average, the throughput of a method like VISTA is only **58.1%** of the vanilla models.

Table 4: **Throughput and Latency Comparison on Three LVLMs.** Measurements are conducted using greedy decoding to evaluate the computational overhead of different methods on the CHAIR benchmark. Throughput is measured in tokens per second (higher is better), and latency is the time in milliseconds per token (lower is better).

| **Method** | LLaVA-1.5 (Liu et al., 2024a) | | Idefics2 (Laurençon et al., 2024) | | InstructBLIP (Dai et al., 2023) | |
|---|---|---|---|---|---|---|
| | ms/token ↓ | token/s ↑ | ms/token ↓ | token/s ↑ | ms/token ↓ | token/s ↑ |
| Vanilla | 17.6 | 56.7 | 20.9 | 47.8 | 16.1 | 62.3 |
| VCD (Leng et al., 2023) | 33.2 | 30.1 | - | - | - | - |
| PAI (Liu et al., 2024b) | 33.9 | 29.5 | - | - | - | - |
| VISTA (Li et al., 2025) | 27.7 | 36.1 | 31.4 | 31.9 | 34.6 | 28.9 |
| **RUDDER-Beta** (Ours) | 18.2 | 54.9 | 21.8 | 45.8 | 16.8 | 59.5 |
| **RUDDER-Add** (Ours) | 17.9 | 55.8 | 21.5 | 46.5 | 16.4 | 60.8 |

## 4.5 ABLATION STUDIES

We conduct an ablation study on Idefics2 using the CHAIR benchmark to analyze the key hyper-parameters: injection layer $L$, maximum steering strength $\alpha_{\max}$ and the gate sensitivity $k$. We first identify the optimal intervention layer, finding Layer 28 is the most effective for the Idefics2 model, as shown in Figure 3a. Focusing on this layer, we then tune the hyperparameters $\alpha_{\max}$ and $k$. The heatmaps in Figures 3b through 3d reveal a core trade-off: increasing the steering strength ($\alpha_{\max}$)

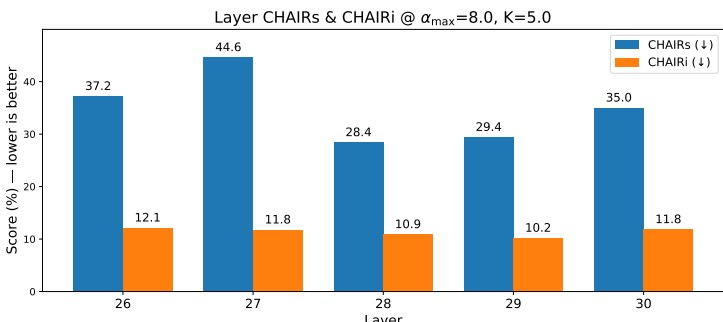

(a) Layer ablation at ($\alpha_{\max} = 8.0$, $k = 5.0$). Mid–late layers ($L \approx 28$–30) are most effective; $L=28$ yields a strong reduction.

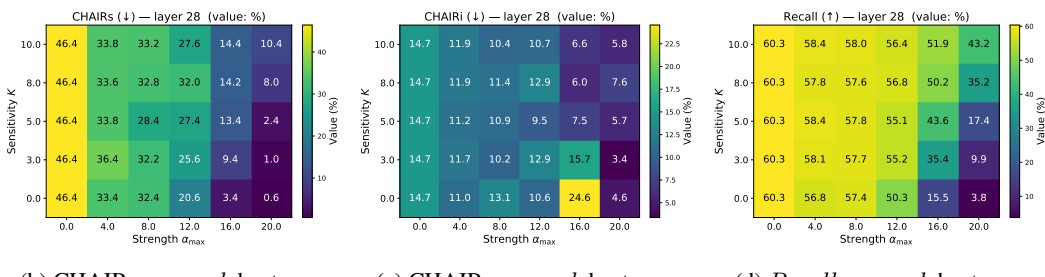

(b) CHAIR$_S$ $\alpha_{\max} \times k$ heatmap.    (c) CHAIR$_i$ $\alpha_{\max} \times k$ heatmap.    (d) $Recall$ $\alpha_{\max} \times k$ heatmap.

Figure 3: **Ablation study of RUDDER's hyperparameters on the Idefics2** (Laurençon et al., 2024) model. **(a)** The bar plot shows the impact of the intervention layer L. **(b-d)** The heatmaps analyze the trade-off between steering strength $\alpha_{\max}$ and gate sensitivity $k$, showing their effect on CHAIR scores and recall.

effectively reduces CHAIR scores but at the cost of lower recall. The gate sensitivity $k$, does not exhibit a simple linear trend; instead, it plays a crucial modulating role in this trade-off. Ultimately, we find that the best balance for Idefics2 is achieved with $\alpha_{\max} = 8.0$ and $k = 5.0$. Ablation results for other models are presented in Appendix B.2.

### 4.6 CASE STUDY

Qualitative analysis in Appendix B.3 demonstrates RUDDER's effectiveness. The case studies show that RUDDER not only eliminates object hallucinations present in the vanilla model's outputs but also produces more conservative content. By avoiding the vanilla model's confident yet incorrect assertions, RUDDER enhances the model's overall reliability.

## 5 CONCLUSION AND LIMITATIONS

In this work, we introduce RUDDER, a low-overhead inference-time intervention framework that mitigates LVLMs hallucination using two key innovations: the zero-cost **CARD vector**, which extracts a per-sample visual evidence from the model's own residual updates, and the adaptive **Beta Gate**, which applies a corrective signal with principled, token-wise strength. Experiments confirm RUDDER achieves state-of-the-art comparable performance on benchmarks like CHAIR and POPE with negligible computational overhead, resolving the common efficacy-efficiency trade-off. RUD-DER presents a practical and effective solution for enhancing the reliability of LVLMs in real-world settings. RUDDER's primary limitation is its sensitivity to hyperparameters, which must be tuned for each model architecture. Future work could focus on automated hyperparameter optimization to improve its robustness and ease of deployment.

## ETHIC AND REPRODUCIBILITY STATEMENT

Our research aims to improve the reliability of LVLMs by mitigating object hallucination. By avoiding the extra forward passes required by many alternative methods, our approach offers a more sustainable path to enhancing model safety.

All experiments are conducted on publicly available benchmarks. Our code is available as open source at the link provided in the Abstract. For our comparative analysis, we reproduced all baseline results using the authors' publicly available code whenever possible. The only exception was VISTA (Li et al., 2025) on the Idefics2 (Laurençon et al., 2024), which we implemented ourselves based on its original code.

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

## A  ADDITIONAL ILLUSTRATION ON THE METHODOLOGY

### A.1  RUDDER ALGORITHM

Here we present the pseudo-code for RUDDER illustrated in Sec. 3.

### A.2  FROM NAÏVE BAYES TO THE BAYESIAN GATE

**Problem setup.** At each decoding step $t$, we want a scalar gate $g_t \in (0, 1)$ that reflects how much the current token should be nudged toward the visual evidence direction $\mathbf{v}_{\text{CARD}}$. Let the alignment statistic be $s_t = \cos(\mathbf{h}_t, \mathbf{v}_{\text{CARD}}) \in [-1, 1]$.

---

**Algorithm 1** RUDDER (single-pass, test-time steering; fixed target layer $\ell$)

---

**Require:** Model $M$; image $x_{\text{img}}$, text $x_{\text{text}}$; fixed layer $\ell$; hyperparams $(\alpha_{\max}, k, c, g_{\min}, g_{\max})$
1: $\mathcal{T}_{\text{pre}} \leftarrow \text{TokenizePrefill}(x_{\text{img}}, x_{\text{text}})$           ▷ image + prompt tokens
2: **Prefill:** run $M$ once (read-only hook at layer $\ell$) to build KV cache and cache $\{\mathbf{A}_i^\ell\}_{i \in \mathcal{T}_{\text{pre}}}$
3: $\Delta_i^\ell \leftarrow \mathbf{A}_i^\ell$ **by** Eq. 1           ▷ pre-norm: residual update equals SA output
4: $\mathbf{v}_{\text{CARD}} \leftarrow$ **by** Eq. 2 (Pool $\rightarrow L_2$-Normalize over $\{\Delta_i^\ell\}_{i \in \mathcal{T}_{\text{pre}}}$)
5: **Decode:** for $t = 1, 2, \ldots$           ▷ auto-regressive generation
6:     $s_t \leftarrow \cos\big(\mathbf{h}_{\ell,t}, \mathbf{v}_{\text{CARD}}^\ell\big)$
7:     $(\alpha_t, \beta_t, g_{\ell,t}) \leftarrow$ **by** Eq. 3;     $g_{\ell,t} \leftarrow \text{clip}(g_{\ell,t}, g_{\min}, g_{\max})$
8:     $\mathbf{v}_t^{\text{steer}} \leftarrow \alpha_{\max} g_t \mathbf{v}_{\text{CARD}}$ **by** Eq. 4
9:     $\mathbf{h}_{\ell,t}^{new} \leftarrow \big(\mathbf{h}_{\ell,t} + \text{SA}^{(\ell)}(\mathbf{h}_{\ell,t})\big) + \mathbf{1}[t \in \mathcal{T}_{\text{ans}}] \cdot \mathbf{v}_t^{\text{steer}}$    ▷ post-SA residual add; **answer span only**
10:     emit next token

---

**Naïve Bayes view (posterior as a gate).** Introduce a latent Bernoulli variable $Z_t \in \{0, 1\}$ indicating whether the token is visually grounded ($Z_t = 1$) or at risk of drifting ($Z_t = 0$). We use the posterior mean $g_t = \mathbb{E}[Z_t \mid s_t]$ as a *continuous* gate (rather than a hard on/off decision).

**Beta–Bernoulli conjugacy with "soft counts".** With a Beta prior $\text{Beta}(\alpha_t, \beta_t)$ on $Z_t$, the posterior mean is

$$g_t = \frac{\alpha_t}{\alpha_t + \beta_t}.$$

We map the alignment $s_t$ to *positive pseudo-counts* via a smooth, monotone transform:

$$\alpha_t = \text{softplus}(k\, s_t + c), \qquad \beta_t = \text{softplus}(-k\, s_t + c),$$

where $k$ controls sensitivity and $c$ controls concentration/bias. The softplus ensures strictly positive, numerically stable "counts".

**Properties (useful for calibration).** The resulting $g_t$ is monotone in $s_t$, symmetric $g(-s) = 1 - g(s)$, and bounded in $(0, 1)$. Around $s = 0$, the slope

$$\left.\frac{\partial g}{\partial s}\right|_{s=0} = \frac{k\, \sigma(c)}{2\, \text{softplus}(c)}, \quad \sigma(x) = \frac{1}{1 + e^{-x}},$$

gives a handy knob to set how fast the gate reacts to alignment changes.

**Stability: clamping and optional per-token cap.** For robustness we clamp $g_t \leftarrow \text{clip}(g_t; g_{\min}, g_{\max})$ to avoid both shutting off ($g_t \rightarrow 0$) and saturating ($g_t \rightarrow 1$). Optionally, we enforce a per-token norm cap $\tau$:

$$\big\|\alpha_{\max} g_t\, \widehat{\mathbf{v}}_{\text{CARD}}\big\|_2 \leq \tau, \qquad \widehat{\mathbf{v}} = \mathbf{v}/\|\mathbf{v}\|_2,$$

which further prevents rare spikes when hidden-state norms vary.

**Final update (matches Algorithm 1).**

$$\mathbf{v}_t^{\text{steer}} = \underbrace{\big(\alpha_{\max} g_t\big)}_{\text{adaptive strength}} \mathbf{v}_{\text{CARD}} \tag{6}$$

$$\mathbf{h}_{l,t}^{\text{new}} = \big(\mathbf{h}_{l,t} + \text{SA}(\mathbf{h}_{l,t})\big) + \mathbf{v}_t^{\text{steer}} \tag{7}$$

We apply this only on the answer span using the mask $m_t$ as in Algorithm 1.

**Implementation notes.**

- We compute $s_t$ with L2-normalized $\mathbf{h}_t$ and $\mathbf{v}_{\text{CARD}}$ (cosine similarity).
- $g_t$ is clamped to $[g_{\min}, g_{\max}]$; the global scale $\alpha_{\max}$ controls the maximal push.
- $\mathbf{v}_{\text{CARD}}$ is extracted once during the mandatory prefill pass (zero extra forwards).

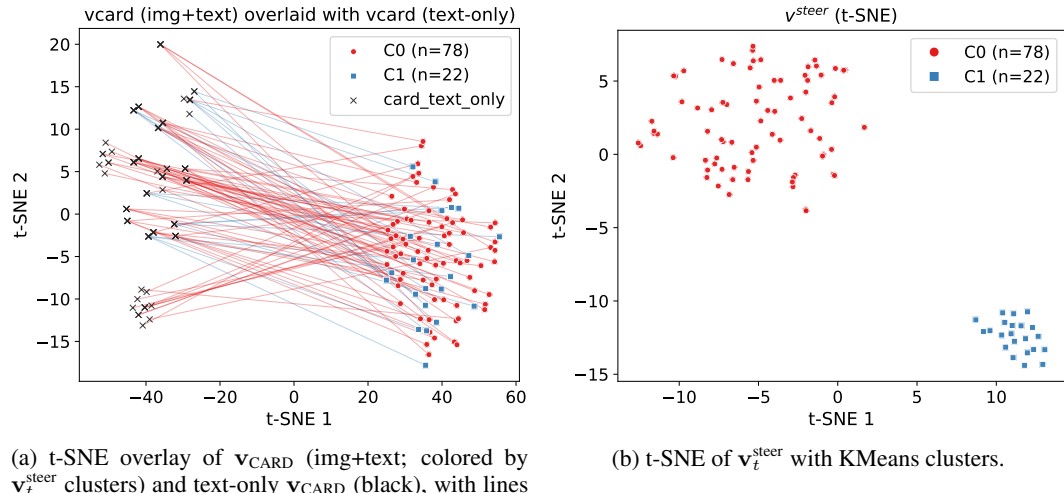

(a) t-SNE overlay of $\mathbf{v}_{\mathrm{CARD}}$ (img+text; colored by $\mathbf{v}_t^{\mathrm{steer}}$ clusters) and text-only $\mathbf{v}_{\mathrm{CARD}}$ (black), with lines linking paired samples.

(b) t-SNE of $\mathbf{v}_t^{\mathrm{steer}}$ with KMeans clusters.

Figure 4: Structure in steering space (b) and its sample-wise projection to $\mathbf{v}_{\mathrm{CARD}}$ (a).

**Practical calibration recipe.** Choose $c$ to set the overall smoothness (typical $c \in [0.5, 1.5]$), then increase $k$ until $g_t$ becomes sufficiently responsive on a small dev set. Finally tune $\alpha_{\mathrm{max}}$ and $[g_{\mathrm{min}}, g_{\mathrm{max}}]$ for stability/strength trade-offs.

**Complexity.** The gate requires only light-weight vector ops during decoding and reuses the prefill to compute $\mathbf{v}_{\mathrm{CARD}}$. Hence no extra forward pass compared to vanilla generation.

### A.3 CLARIFICATION

RUDDER maintains the same inference efficiency as the original model, requiring no extra forward passes. The CARD vector is extracted opportunistically during the mandatory prefill pass, and the steering is applied within each step of the subsequent decoding pass.

By no extra forward pass we mean no additional model.forward invocations beyond the vanilla prefill and decode; our overhead comes only from cheap per-token vector operations implemented via hooks.

### A.4 VISUALIZATION AND QUANTIFICATION OF THE GEOMETRY OF $\mathbf{v}_{\mathrm{CARD}}$ AND $\mathbf{v}_t^{\mathrm{STEER}}$

**Setup and link to motivation.** Large VLMs fuse vision and text via self-attention; the residual update of this sublayer thus captures the *net* impact of visual context on token representations. Motivated by this, we aggregate the prefill-phase residual updates to obtain a per-sample direction $\mathbf{v}_{\mathrm{CARD}} \in \mathbb{R}^d$ and define its steering counterpart $\mathbf{v}_t^{\mathrm{steer}} = (\alpha_{\mathrm{max}} \overline{\mathrm{gate}}) \mathbf{v}_{\mathrm{CARD}}$ (Sec. 3). For each image we export (i) $\mathbf{v}_{\mathrm{CARD}}$ (image+prompt) and its text-only variant, and (ii) $\mathbf{v}_t^{\mathrm{steer}}$. We reduce vectors by PCA ($k=50$) then t-SNE (default perplexity unless noted), and cluster the steering space with KMeans (best silhouette over $K \in \{2, \ldots, 10\}$). Figure 4 shows two key views: a *paired* overlay of image+text *vs.* text-only $\mathbf{v}_{\mathrm{CARD}}$ with one-to-one lines, and the clustered t-SNE of $\mathbf{v}_t^{\mathrm{steer}}$. The overlay reveals systematic sample-wise *rotations* from the language prior (text-only) to the image-conditioned direction, and these rotations point toward coherent steering clusters—visual evidence is therefore *directional* rather than noise, directly supporting our motivation.

**Quantifying directional structure.** We quantify two effects central to our hypothesis: **(i) Rotation from text-only to image-conditioned CARD:** $\Delta\theta = \arccos\langle\mathbf{v}_{\mathrm{text}}, \mathbf{v}_{\mathrm{img+txt}}\rangle$ shows a tight distribution around $\sim40°$ (mean $\approx40.5°$, median $\approx40.4°$), indicating a consistent, non-trivial visual-induced rotation rather than random drift (Fig. 5a); this effect persists across steering clusters (Fig. 5c). **(ii) Alignment gain w.r.t. steering:** $\langle\mathbf{v}_{\mathrm{img+txt}}, \mathbf{v}^{\mathrm{steer}}\rangle - \langle\mathbf{v}_{\mathrm{text}}, \mathbf{v}^{\mathrm{steer}}\rangle$ is positive on average (mean $\approx0.239$, median $\approx0.238$) and remains positive across clusters (Figs. 5b,d), showing that

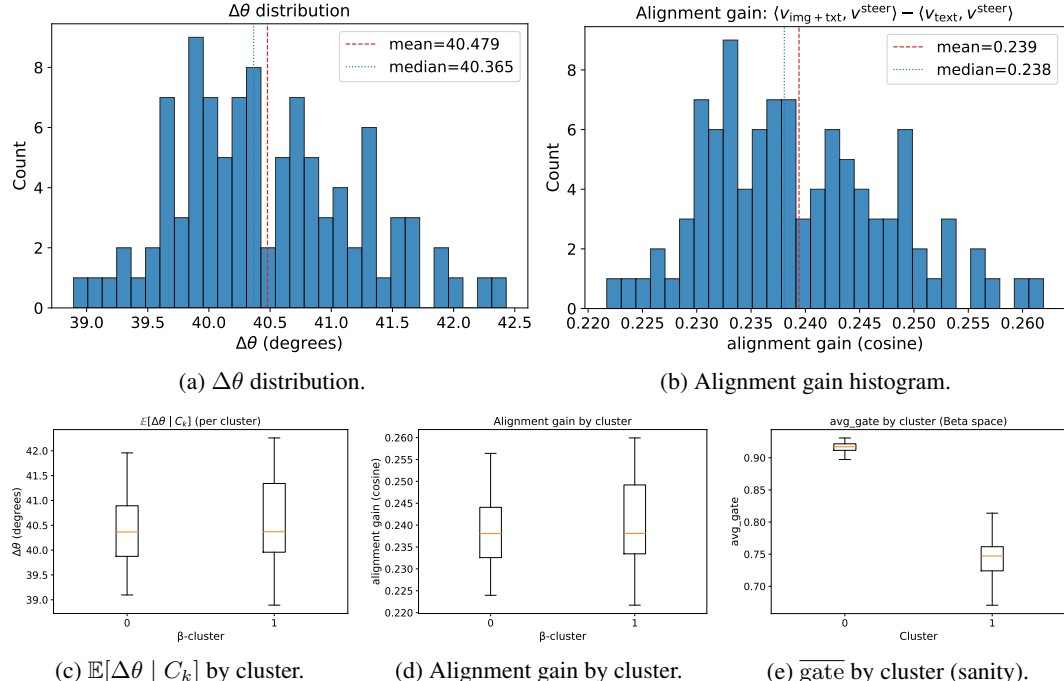

(a) $\Delta\theta$ distribution.

(b) Alignment gain histogram.

(c) $\mathbb{E}[\Delta\theta \mid C_k]$ by cluster.

(d) Alignment gain by cluster.

(e) $\overline{\text{gate}}$ by cluster (sanity).

Figure 5: Directional evidence with reflowed layout. (a) Consistent $\mathbf{v}_{\text{text}} \rightarrow \mathbf{v}_{\text{img+txt}}$ rotation; (b) positive alignment gain to $\mathbf{v}^{\text{steer}}$; (c,d) cluster-wise stability; (e) systematic gate differences.

image-conditioned $\mathbf{v}_{\text{CARD}}$ is *closer* to the actual steering geometry used by the $\beta$-gate. Together, these results substantiate our motivation: aggregating self-attention residual updates yields a robust sample-specific visual-evidence direction that aligns with the downstream steering mechanism.

**Notes.** Silhouette scores are typically higher for $\mathbf{v}_t^{\text{steer}}$ than for $\mathbf{v}_{\text{CARD}}$, consistent with the gate organizing/scaling directions across samples. We emphasize that t-SNE primarily supports *local* neighborhood interpretation; all scalar statistics are computed in the original vector spaces.

# B  ADDITIONAL EXPERIMENT

## B.1  LLaVA INTERNAL DYNAMIC ANALYSIS

As mentioned in Section 3.2, our method is guided by an analysis of the internal dynamics of LLaVA-1.5 (Liu et al., 2024a), with key findings illustrated in Figure 6. By examining the residual update vector from the self-attention module at each layer, we identified two properties that informed our intervention strategy:

**Intervention Leverage Peaks in Late Layers.** We can find that the magnitude of the residual update, which represents the "leverage" an intervention can have, is not uniform. We found that its strength grows with model depth, peaking in the late decoder blocks (approx. layers 26-32). This indicates that interventions in these layers have the greatest potential to influence the model's final output (Figure 6a, 6c).

**Directional Coherence Collapses in Late Layers.** While late layers offer the most leverage, the directional coherence of their update vectors collapses after approximately layer 21 (Figure 6b). Coherence is moderate only in the early-to-mid layers. This suggests that applying a fixed, global steering vector in the high-leverage late layers is suboptimal, as the intervention may be misaligned with the model's unstable internal state.

## B.2 ADDITIONAL ABLATION RESULTS

We provide supplementary abalation results on LLaVA-1.5 and InstructBlip, as shown in Figure 11 and Figure 12. These analyses complement the main ablation study conducted on Idefics2 in Section 4.4.

The results for both models confirm the same core trends observed with Idefics2. Specifically, the heatmaps reveal a consistent trade-off between hallucination mitigation and recall. As the steering strength $\alpha_{max}$ increases, both CHAIRS and CHAIRI scores improve (decrease), but this is often accompanied by a drop in recall. The gate sensitivity parameter, $k$, plays a similar, non-linear modulating role in this balance. While the general trade-off is consistent, the optimal hyperparameter values vary by model architecture, highlighting the need for model-specific tuning.

## B.3 CASE STUDY

To provide a qualitative illustration of our method's real-world performance, we present a series of case studies in Figures 7, 8, 9 and 10. From these examples, we can find that RUDDER is highly effective at eliminating object-level hallucinations. It successfully removes entirely non-existent objects from the captions (e.g., a hallucinated "second person" or "cup"), and corrects misidentified objects (e.g., correctly identifying "skis" instead of "snowboard"). Moreover, RUDDER's outputs are not only more factually accurate but also more semantically cautious. The corrected descriptions often adopt more conservative language, such as using phrases like "appears to be", "may be", or "suggesting that". By replacing the baseline models' confident yet incorrect assertions with more grounded and appropriately qualified statements, RUDDER significantly enhances the overall reliability and trustworthiness of the generated text.

# C LLMS USAGE STATEMENT

Generative AI has been utilized to enhance the writing and to assist with coding tasks.

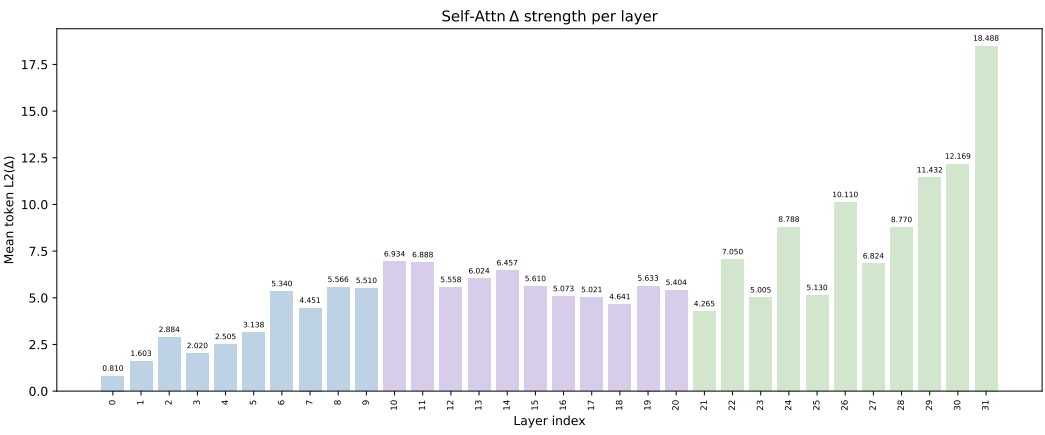

(a) Absolute strength of the update vector ($\|\delta^l\|$) across layers.

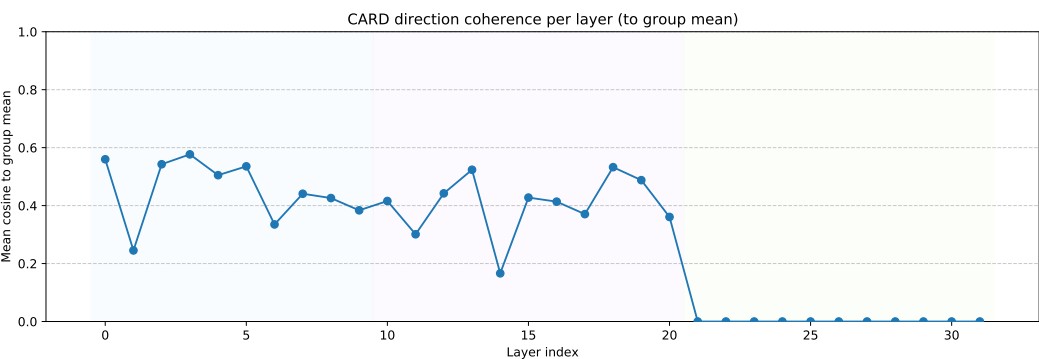

(b) Directional coherence within layer groups.

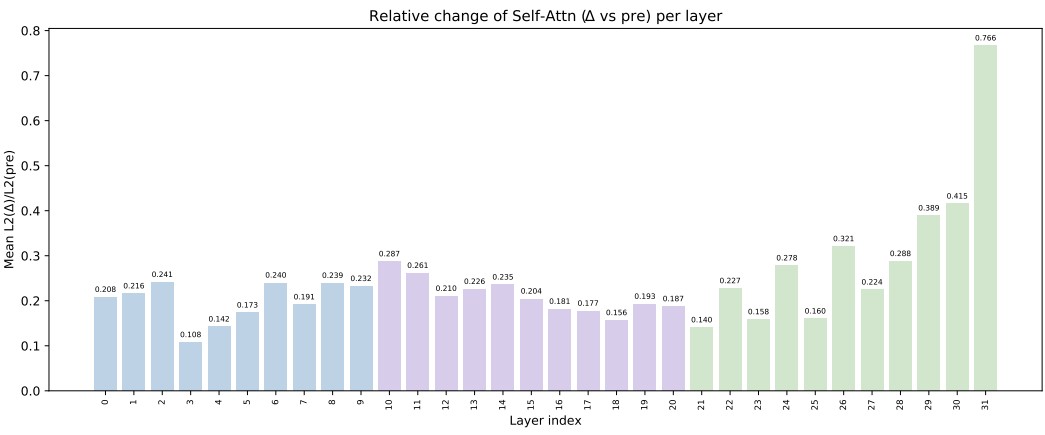

(c) Relative strength of the update vector ($\|\delta^l\|/\|h_{\text{pre}}^l\|$).

Figure 6: Analysis of the internal dynamics of LLaVA-1.5 (Liu et al., 2024a). (a, c) Both the absolute and relative strength of self-attention updates peak in the middle-to-late layers, which identify a "computational core". (b) Mid-late layers show significantly lower directional coherence than other regions.

**Baseline:**
"In the image, a man is standing in front of a motorcycle with his legs crossed. He is wearing a helmet and safety gear, possibly indicating that he is the motorcycle owner or rider. The motorcycle is prominently placed in the scene, occupying a significant portion of the image. The man seems to be posing for the photo and showcasing his motorcycle.

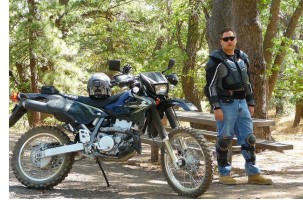

**RUDDER:**
"In the image, a man stands close to a parked black dirt bike and sidewalk benches. He is wearing a blue outfit and appears to be enjoying the environment around him. The dirt bike is parked prominently in front of him, taking up most of the frame. There are benches placed further back from the man, creating a natural seating area in the scene."

**Baseline:**
"The image captures a man standing in a room, fully engaged in playing a video game on a TV. He is using a Nintendo Wii controller to control his gameplay. In addition to the man, there is another person located to the far left of the image. There are also several books scattered across the room, along with a cup placed on a surface. A potted plant can be found towards the right side of the room, adding a touch of greenery to the space."

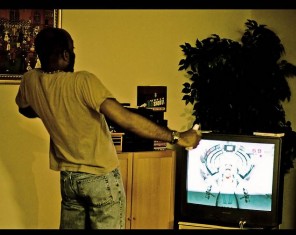

**RUDDER:**
"In the image, a man is actively playing a video game on his living room. TV setup stands outstanding. He is wearing jeans and appears to be dancing while swinging a controller or using a motion device that allows him to physically participate in the gameplay. There are multiple books visible on a table or shelf, suggesting that this living area may be used for studying or enjoying various forms of reading material. Additionally, a potted plant is present in the space adding a touch of greenery to the setting."

Figure 7: Case study. Hallucinated contents generated by the vanilla LLaVA-1.5 are marked in red, while the factual contents from our method are colored with blue.

**Baseline:**
"The image features a small, clean kitchen with a white countertop and a stove top oven. The kitchen is equipped with a sink and a microwave, and there is a toilet visible in the background. The room also has a dining table and a few chairs placed around it. In addition to the main kitchen appliances, there are a few other items in the room. "

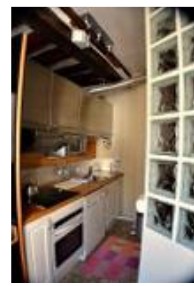

**RUDDER:**
"The image features a clean and tidy kitchen with a variety of appliances and utensils. There is a sink positioned towards the center of the room, with a microwave oven placed above it. A refrigerator can be seen on the left side of the room, while an oven is located towards the bottom left corner.

**Baseline:**
"The image features a cozy living room with a couch and a chair. On the couch, there are two cats lying down, one on the left side and the other on the right side. They appear to be relaxed and enjoying their time together. In addition to the cats, there are two books placed on the couch, one near the left cat and the other near the right cat. The scene gives off a warm and comfortable atmosphere, with the cats and books adding to the overall ambiance of the living room."

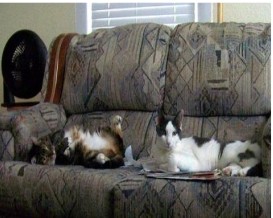

**RUDDER:**
"The image features two cats lying down on a couch, enjoying each other' sierp company. One cat is positioned towards the left side of the couch, while the other cat is laying down further to the right side. They both seem to be resting comfortably on newspapers that are spread out on the couch. The scene creates a cozy atmosphere with the cats sharing their space on the couch."

**Baseline:**
"The image features a wooden table with a variety of kitchen utensils and items. There are several knives, including a large knife and a smaller one, placed on the table. A spoon is also visible, along with a couple of spoons and a fork. In addition to the utensils, there are multiple bottles on the table, with one being larger and positioned towards the left side, and the others being smaller and scattered around the table. A cup can be seen on the right side of the table, and a bowl is placed in the middle. table."

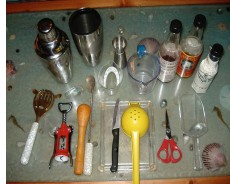

**RUDDER:**
"The image features a tabletop with a variety of items laid out on it. There are several spices neatly arranged in a line. Additionally, a number of spoons are positioned around the tabletop, some pointing upwards and others pointing downwards. There are also several bottles dispersed across the tabletop, adding to the assortment of items present. The scene gives off an impression of a well-organized and functional space for preparing meals or engaging in culinary activities."

**Baseline:**
"The image features a red fire hydrant situated on a sidewalk next to a building. The fire hydrant is prominently placed in the scene, and it appears to be the main focus of the image. The sidewalk is located near a street, and there is a truck parked on the street, slightly to the left of the fire hydrant."

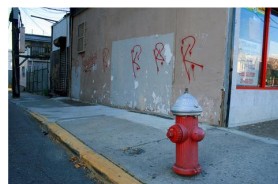

**RUDDER:**
"The image features a red fire hydrant prominently positioned on the sidewalk near a building wall. The fire hydrant is sitting on top of a sidewalk curbing, making it easily accessible for emergency services. In addition to the fire hydrant, there are several graffiti tags on the building wall behind it. These tags add an urban touch to the scene and provide contrasting visual interest against the red fire hydrant."

Figure 8: Case study. Hallucinated contents generated by the vanilla LLaVA-1.5 are marked in red, while the factual contents from our method are colored with blue.

**Baseline:**
"The image features a man wearing a jacket and a cap, standing on a snowy hill. He appears to be posing for a photo while holding his snowboard. The man has his hands close to his chest. The snowy terrain surrounds the man, and in the background, there are additional trees scattered across the area. A second person is visible in the background, standing further to the right. "

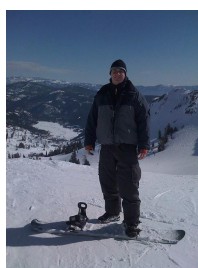

**RUDDER:**
"The image features a man posing on top of a snow-covered ski slope during winter. He is wearing snow gear and is standing confidently on skis. The scene appears to be captured during the winter season, with the mountainous setting serving as a striking background."

**Baseline:**
"The image features a busy city street with several police officers patrolling the area on their motorcycles. There are three police officers prominently visible in the scene, with one officer positioned slightly to the left of the center, another officer slightly to the right of the center, and the third officer slightly to the right of the rightmost officer. Multiple cars are parked or driving down the street, including one car parked on the left side of the image, another car parked further down the street on the right side, and several cars driving down the roadway. Additionally, there are two handbags visible in the scene one near the leftmost officer and the other slightly to the right of the center officer."

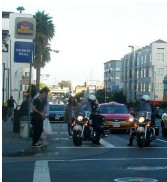

**RUDDER:**
"The image depicts a busy city street filled with various vehicles and pedestrians. There are two police officers on motorcycles, one on the left side and the other on the right side of the street. They appear to be directing traffic or monitoring the area. In addition to the motorcycles, there are several cars and a truck on the street. Some cars are parked, while others seem to be in motion. A few pedestrians can be seen walking along the sidewalk, with one person carrying a handbag. Overall, the scene captures the hustle and bustle of city life, with the police officers playing a crucial role in maintaining order and safety."

**Baseline:**
" The image captures a lively concert scene with a crowd of people gathered around a stage. A man is standing on the stage, singing into a microphone, while a woman is also present on the stage, possibly accompanying the singer. There are several people in the audience, some of them holding cell phones, likely capturing the moment or sharing the experience with others. The atmosphere appears to be energetic and engaging, with the performers and the audience enjoying the concert together."

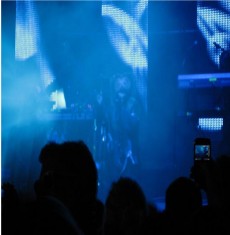

**RUDDER:**
"The image features a live concert event taking place indoors. At the center of attention is a person singing into a microphone while a band plays instruments behind them. There are several other people watching the performance from various positions around the room. Some of these individuals are standing close to each other, while others are spread out across the venue. In addition to the live performance setup, there are multiple cell phones being used by concertgoers to capture memories of the event. "

**Baseline:**
" The image features a black cat sitting in a bathroom sink, surrounded by various toiletries. The sink is filled with toothbrushes, some of which are placed near the cat, while others are scattered around the sink. There are also multiple bottles in the scene, with some located near the cat and others placed further away. The cat appears to be relaxed and comfortable in the sink, surrounded by the bathroom items."

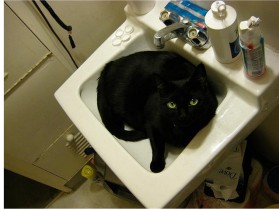

**RUDDER:**
"The image features a black cat comfortably lying down inside a bathroom sink. The sink occupies most of the scene, with the cat occupying its central space. The cat appears to be enjoying its time in the sink, possibly finding it cozy or cooler than its surroundings."

Figure 9: Case study. Hallucinated contents generated by the vanilla Idefics2 are marked in red, while the factual contents from our method are colored with blue.

**Baseline:**
" There are four plates with different types of cakes and pastries, each with a unique flavor and design. The desserts are arranged in a way that they are easily accessible for the diners. The table is set with utensils, including a fork and a knife, and a cup is also present. The dining area is well-lit, providing a pleasant atmosphere for enjoying the delicious treats."

**Baseline:**
" He is looking upwards, possibly in a contemplative or thoughtful manner. The room appears to be a dining area or a gathering place, with a table and chairs visible in the background. The man's facial expression and body posture suggest a sense of seriousness or concentration "

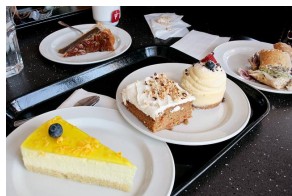

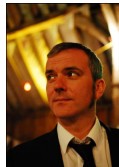

**RUDDER:**
" There are four plates on the table, each containing a different dessert. The dessert selection includes a slice of cake, a slice of cheesecake, a slice of lemon cake, and a slice of chocolate cake. The dessert plates are arranged in a way that allows for a variety of flavors to be enjoyed."

**RUDDER:**
"He is looking upwards, possibly at a person or object in the room. The man is the focal point of the scene, and his presence is quite prominent. There are two lights in the room, one on the left side and another on the right side. The man is the center of the scene, and his presence is quite prominent."

**Baseline:**
" Two men and a woman present, all wearing different clothing. One man is holding a kite, while the other man is holding a bottle. The woman is standing nearby, possibly assisting with the kite or enjoying the activity. "

**Baseline:**
" The baby is wearing a white outfit, and the man is smiling at the camera. The baby is positioned close to the man's face, making it a heartwarming moment captured in the photo. There are several other people in the background, including a man and a woman, both standing in the same room. The man and woman are positioned on opposite sides of the room, with the man holding the baby and the woman standing in the background "

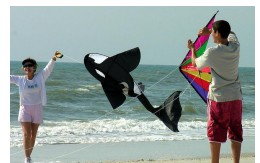

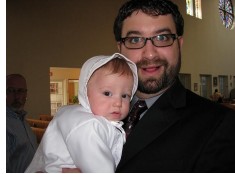

**RUDDER:**
" There are two kites in the air, one of a kite shaped like a killer whale. The people are having fun and flying the kites, possibly enjoying the day together."

**RUDDER:**
" The baby is wearing a white outfit, and the man is smiling at the camera. There are two other people in the room, one on the left side and another on the right side. The room appears to be a church, with a cross visible in the background. The man and the baby are the focal point of the scene."

Figure 10: Case study. Hallucinated contents generated by the vanilla InstructBlip are marked in red, while the factual contents from our method are colored with blue.

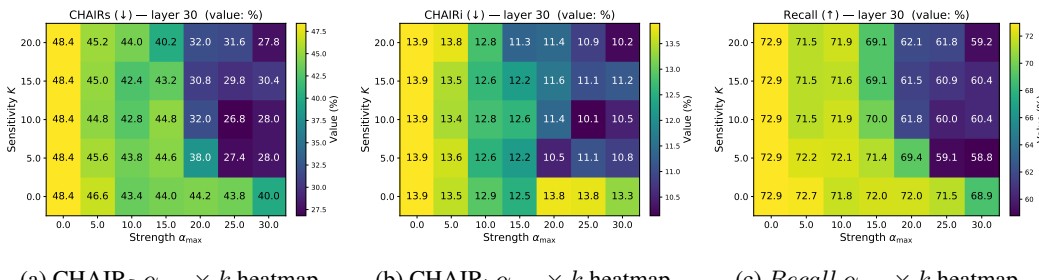

(a) CHAIR$_S$ $\alpha_{max} \times k$ heatmap.  (b) CHAIR$_i$ $\alpha_{max} \times k$ heatmap.  (c) *Recall* $\alpha_{max} \times k$ heatmap.

Figure 11: Ablation matrices for RUDDER on LLaVA-1.5 (Liu et al., 2024a)

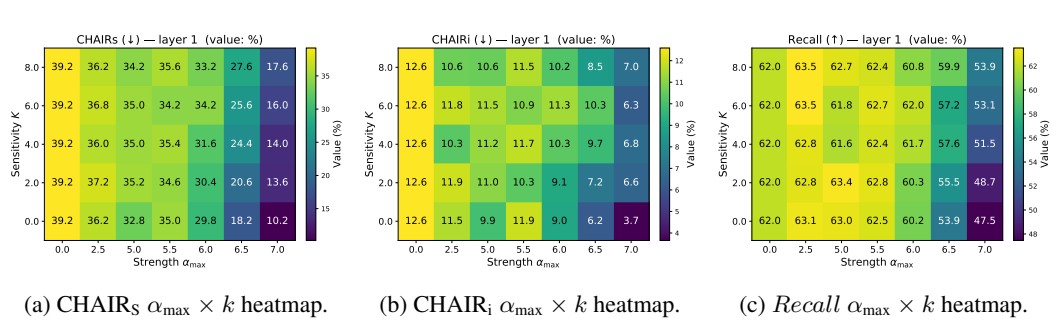

(a) CHAIR$_S$ $\alpha_{max} \times k$ heatmap.  (b) CHAIR$_i$ $\alpha_{max} \times k$ heatmap.  (c) $Recall$ $\alpha_{max} \times k$ heatmap.

Figure 12: Ablation matrices for RUDDER on InstructBlip (Dai et al., 2023)

