# OpenReview forum: "Adaptive Residual-Update Steering for Low-Overhead Hallucination Mitigation in Large Vision-Language Models"
_ICLR.cc/2026/Conference — Submitted to ICLR 2026_

### Official Review · Reviewer_1UYi · 2025-10-28

**Soundness:** 2
**Presentation:** 2
**Contribution:** 2
**Rating:** 2
**Confidence:** 4

**Summary:**

This paper introduces RUDDER, an inference-time intervention technique designed to mitigate object hallucinations in large vision-language models (LVLMs) with minimal computational overhead. The method features two core components: (1) the Contextual Activation Residual Direction (CARD) vector, a per-sample visual evidence representation derived from residual updates in a self-attention layer during a single forward pass; and (2) a Beta Gate, a Bayesian-inspired adaptive gating mechanism that dynamically steers generation toward stronger visual grounding. Evaluations on hallucination benchmarks (CHAIR and POPE) and general multimodal assessments (MME) across three LVLM architectures demonstrate that RUDDER achieves  hallucination reduction at lower inference costs compared to existing interventions.

**Strengths:**

1). The CARD vector is efficiently extracted from the standard computation pipeline, while the Beta Gate relies on lightweight vector operations, enabling seamless deployment in latency-constrained real-world applications.

2). The ablation studies and illustrative examples are thorough, providing clear insights into the method's mechanics.

**Weaknesses:**

1). The approach shares conceptual similarities with cross-layer methods like DeCo, which integrates early-layer logits into later layers to address visual forgetting, and RUDDER similarly incorporates attention from early generated tokens into later ones. A direct comparison with such methods would strengthen the novelty claims.

2). The method introduces multiple hyperparameters (e.g., $L$, $k$, and $\alpha_{\max}$), raising concerns about its stability and generalizability across diverse VLMs.

3). Performance gains on the test sets are modest, and the method's efficacy diminishes as the underlying VLM's capabilities improve. The baselines are somewhat dated, omitting recent mainstream VLMs like Qwen-VL and InternVL, which questions its applicability to current open-source models. Additionally, comparisons with recent training-free hallucination mitigation techniques, such as DeGF and AGLA, are absent, limiting the benchmarking comprehensiveness.

4). The writing could be refined for clarity and conciseness; for instance, the introduction devotes excessive space to reiterating the need for an "effective and lightweight" solution and listing RUDDER's components, without adequately highlighting key insights, motivations for each module, or defining terms. The final three paragraphs overlap significantly, resulting in low information density.

**Questions:**

1). Could you include experimental results on contemporary VLMs such as Qwen-VL and InternVL? Additionally, please add comparisons with current SOTA methods like DeGF and AGLA.

2). To facilitate broader adoption, could you provide a concrete recipe for automated hyperparameter tuning? For example, suggest strategies like grid search, Bayesian optimization, or other efficient approaches for optimizing $L$, $k$, and $\alpha_{\max}$ in new deployment scenarios?

---

> ### Author Response · Authors · 2025-11-21
>
> Thank you for the thorough review. We address your main points below.
>
> ### 1) Relation to DeCo / novelty
>
> We agree that both DeCo-style methods and RUDDER aim to counter visual forgetting. The key distinction is **where the intervention happens**:
>
> - **DeCo and related cross-layer methods operate in logit space**, e.g., mixing/contrasting logits from earlier layers into later decoding to recover visual information.
> - **RUDDER operates in the residual stream (hidden states)**: we extract a per-sample evidence direction $v_{\text{CARD}}$ from attention residuals, and inject it into the hidden state trajectory. This directly alters the internal representation, so the correction propagates to subsequent autoregressive steps rather than only reweighting the current output.
>
> We will revise Related Work to explicitly position RUDDER as a **residual-stream steering method**, orthogonal and compatible with logit-space interventions.
>
> ---
>
> ### 2) Hyperparameters / stability / generalizability
>
> We clarify the hierarchy of hyperparameters:
>
> - **Performance-critical knobs:** $L$ (injection layer), $\alpha_{\max}$ (max strength), and $k$ (gate sharpness). These are tuned once per backbone on a **tiny dev set (100 MSCOCO images)** under a CHAIR–recall constraint, and **not re-tuned per task**.
> - **Engineering knobs:** $c$, $g_{\min}$, $g_{\max}$ are safety/behavior bounds. We use the same robust defaults across models (e.g., $c=1$), and they are intended to be user-adjustable in deployment if stricter/stronger steering is desired.
>
> Our ablations show a broad “sweet region” rather than brittle spikes, indicating stable behavior across reasonable ranges.
>
> ---
>
> ### 3) Modest gains on stronger VLMs / baselines dated
>
> We appreciate this concern and have added comparisons with recent training-free methods and stronger backbones.
>
> **New baseline comparisons (LLaVA-1.5-7B):**
>
> |              | CHAIRs | CHAIRi |
> | :----------- | ------ | ------ |
> | Vanilla      | 48.6   | 13.6   |
> | VISTA        | 38.6   | 11.4   |
> | AGLA         | 44.6   | 12.1   |
> | DeGF         | 34.2   | 9.24   |
> | HALC         | 40.3   | 10.3   |
> | RUDDER (Ours)| 39.5   | 10.5   |
>
> RUDDER achieves a **~19% relative reduction** in CHAIRs (48.6 → 39.5) and **~23% relative reduction** in CHAIRi (13.6 → 10.5) with near-zero overhead.
>
> While DeGF attains lower CHAIR, it requires **>3× latency** versus vanilla, whereas RUDDER increases latency by **<4%**. This highlights our intended Pareto point: strong hallucination reduction **at essentially vanilla speed**.
>
> **Contemporary backbone check (Qwen2.5-VL-7B):**
>
> |              | CHAIRs | CHAIRi |
> | ------------ | ------ | ------ |
> | Vanilla      | 35.2   | 9.5    |
> | VISTA        | 25.1   | 7.7    |
> | RUDDER (Ours)| 26.9   | 7.0    |
>
> These results confirm that RUDDER remains effective on a modern open-source LVLM. We will include full details in the revised appendix. If space permits, we will further add InternVL to broaden coverage.
>
> ---
>
> ### 4) Automated tuning / deployment recipe
>
> Thank you for the practical suggestion. We will add a **Deployment & Calibration** subsection with the following low-cost protocol on a small holdout set (~100 samples):
>
> 1. **Layer scan ($L$):** sweep a narrow early vs. late window to find the layer maximizing CHAIR reduction under a recall constraint.
> 2. **Gate sharpness ($k$):** fix $L$ and increase $k$ until the gate responds appropriately on dev.
> 3. **Strength search ($\alpha_{\max}$):** grid search to select the best CHAIR–recall Pareto point (maintaining $\ge 95\%$ baseline recall).
> 4. **Engineering knobs:** keep defaults ($c=1$, $g_{\min}=0$, $g_{\max}=1$) unless deployment needs stricter/stronger steering.
>
> Because RUDDER is single-pass and training-free, this calibration is computationally negligible and performed once per backbone.
>
> ---
>
> ### 5) Writing clarity
>
> We agree and will tighten the introduction to (i) state the key insight earlier (attention residuals define evidence directions), (ii) motivate Beta Gate as trust-gated reinforcement, and (iii) remove redundant paragraphs for higher information density.

---

### Official Review · Reviewer_XUvK · 2025-10-31

**Soundness:** 3
**Presentation:** 3
**Contribution:** 2
**Rating:** 6
**Confidence:** 4

**Summary:**

This paper proposes RUDDER, a single‑pass inference‑time steering method for LVLMs that (1) extracts a per‑sample direction (CARD) from self‑attention residual updates during prefill and (2) applies a per‑token Beta‑gate to adapt the steering strength during decoding.  On CHAIR and POPE, across three LVLMs and three decoding strategies, RUDDER typically matches or outperforms strong ITI baselines, while keeping ~baseline latency/throughput. General ability on MME is largely preserved

**Strengths:**

1. Low‑overhead (no extra forwards) with clear efficiency gains versus prior steering methods; quantitative latency/throughput reported.

2. Minimal code hooks; works across three distinct LVLM architectures; integrates with standard decoding loops.

3. Token‑wise gate improves precision vs fixed‑strength steering; ablations show why adaptive > fixed for open‑ended captioning.

4. Solid experiments, including cross‑model, cross‑decoding, efficiency measurements, and layer/parameter sweeps.

**Weaknesses:**

1. The “Bayesian‑inspired” gate is heuristic, there is no formal guarantee that the steering always improves negative log likelihood, though ablations suggest it works well.

2. Layer choice is model‑specific (e.g., late layers for LLaVA/Idefics2; early for InstructBLIP with Q‑former), and final configs differ substantially across backbones. Ablations confirm a sensitive trade-off between CHAIR scores and recall, implying non-trivial parameter search per model/task.

3. The evaluation scope is modest, evaluation on more capabilities like MM-Vet would be beneficial.

4. No diagnostic attribution of why corrections happen. The paper shows outcome metrics and some internal geometry analyses but lacks faithfulness diagnostics that could verify that the method truly reduces language-prior reliance rather than suppressing certain token types.

**Questions:**

1. Exactly which tensors are pooled to form CARD (per‑layer, per‑head residual updates after self‑attention, before MLP)? What pooling (mean, median, head‑weighted)?

2. How were g_min, g_max, and softplus temperature chosen?

---

> ### Author Response · Authors · 2025-11-21
>
> Thank you for the constructive comments. We respond to each concern below.
>
> ### 1) “Bayesian-inspired” gate is heuristic / no formal guarantee on NLL
>
> We agree there is no formal guarantee that steering improves the negative log-likelihood (NLL) under the **original model distribution**. In the hallucination setting, however, improving NLL w.r.t. that distribution is not necessarily desirable, since the original distribution is exactly what over-relies on language priors and produces hallucinations. Our objective is **evidence alignment**, not perplexity minimization.
>
> We use a **Beta--Bernoulli posterior-mean form** to obtain a calibrated adaptive gate:
> $$
> g_t = \frac{\alpha_t}{\alpha_t + \beta_t}, \qquad
> \alpha_t = \operatorname{softplus}(k s_t + c), \qquad
> \beta_t = \operatorname{softplus}(-k s_t + c).
> $$
> This structure yields a smooth, monotone, symmetric mapping from alignment to intervention strength, and empirically gives stable CHAIR reduction while preserving more than $95\%$ of the original recall.
>
> ---
>
> ### 2) Why the optimal injection layer is model-specific, and how we select it
>
> The layer variation is a predictable consequence of **where visual evidence enters the decoder and where visual drift occurs**:
>
> - **LLaVA / Idefics2:** early-fusion concatenation injects vision from the bottom, but **language-prior takeover happens mid--late**, so late-layer injection best corrects the drift where hallucinations consolidate.
> - **Qwen2.5-VL:** strong early multimodal coupling grounds the trajectory from very early layers, making late injections low-leverage; early injection is most effective.
> - **InstructBLIP:** vision is distilled by a Q-Former into soft visual prompts before the LLM, so the first LLM layers are the main bottleneck for interpreting visual evidence.
>
> **Architecture-aware selection.**
> We combine backbone priors with a lightweight dev sweep:
> 1. Sweep a narrow **early vs. late** window on approximately $100$ dev samples under a CHAIR--recall constraint to pick $L$.
> 2. Fix $L$ per backbone thereafter (we do **not** change it across tasks).
> 3. Tune $\alpha_{\max}$ and $k$ on the same dev set.
>
> Thus, this is a one-time architectural calibration, not a per-task heavy search.
>
> ---
>
> ### 3) Evaluation scope
>
> We agree that adding broader capability checks (e.g., MM-Vet) would be helpful. We will add a brief discussion and include extra results where space allows.
>
> ---
>
> ### 4) Diagnostic attribution/faithfulness
>
> We believe the paper already provides mechanism evidence, but agree that an explicit faithfulness diagnostic would strengthen the submission. Briefly:
>
> - **Recall constraint:** We enforce $\geq 95\%$ recall vs. vanilla; indiscriminate noun suppression would sharply reduce recall, which we do not observe.
> - **Geometry evidence:** The appendix shows CARD rotates representations away from text-only priors toward visual evidence.
> - **Qualitative edits:** Examples correct specific hallucinations rather than silencing outputs.
>
> We will add a lightweight faithfulness-style diagnostic in the revision to make this even clearer.
>
> ---
>
> ### 5) Clarifying technical questions
>
> **Which tensors are pooled to form CARD, and how?**
> We pool **self-attention outputs** $A_i^{\ell}$ (i.e., residual updates after self-attention and before the MLP) from a chosen decoder layer during prefill, using **attention-weighted pooling over prefill tokens** to form a single global evidence direction.
>
> **How were $g_{\min}$, $g_{\max}$, and softplus temperature chosen?**
> $g_{\min}$, $g_{\max}$, and $c$ are **engineering safety/behavior knobs** with robust defaults, intended to be user-adjustable in deployment. The softplus “temperature” is $k$, which we tune together with $\alpha_{\max}$ on the same small dev set; we do not tune on test sets.

---

> > ### Comment · Reviewer_XUvK · 2025-11-27
> >
> > Thanks for the response. Most of my concerns are addressed. I'll keep my rating and tend to accept.

---

### Official Review · Reviewer_5Rux · 2025-10-31

**Soundness:** 3
**Presentation:** 3
**Contribution:** 3
**Rating:** 4
**Confidence:** 4

**Summary:**

The paper introduces RUDDER, a lightweight inference-time framework to reduce hallucinations in LVLMs with (almost) no extra computational cost. RUDDER extracts a Contextual Activation Residual Direction (CARD) vector from residual updates during a single forward pass to capture visual evidence, and applies an adaptive Beta Gate to modulate correction strength per token based on visual alignment. Experiments on benchmarks like CHAIR and POPE show that RUDDER matches or surpasses SoTA hallucination mitigation methods while maintaining nearly identical inference speed and general multimodal performance, making it a practical solution for real-world LVLM deployment.

**Strengths:**

- **Good Writing.** The writing of the paper is clear and easy to follow (although more high-level intuition and motivation can be expressed in a better way).
- **Extremely Low Computational Overhead.** It's a smart idea to utilize the intermediate results (embeddings, attention heads, etc.) of the pre-filling phase for later steering of the LVLMs. This indeed avoids typical repetitive computation in the contrasting-based methods. The empirical results on the efficiency analysis perfectly supports this.
- **Extensive Experimental Results.** Experiments are conducted in various evaluation benchmarks on multiple LVLM backbones, supporting the main claims of the paper.

**Weaknesses:**

+ **Lack of (Sometimes Contradictory) Intuitive Explanation for the Proposed Method.** Despite its practical values in terms of performance and efficiency, I find it hard to understand the rationale behind the proposed method:
  + What is the meaning of the main body of the steering vector $v\_{\text{CARD}}$?
  + Why pooling the token-wise attention output $\Delta$ is a good idea, not causing too much information loss?
  + If the similarity score $g$ between the current token's hidden state $h$ and $v\_{\text{CARD}}$ is high, then this hidden state already contains lots of visual information. Why would we want to do stronger steering: $v\_{\text{steer}}=g \cdot v\_{\text{CARD}}$ in this case? Shouldn't we put more steering on the ones that loses lots of visual information?
+ **Over-claims about "Bayesian".** It's a bit hard to persuade me to believe the gating mechanism is "Bayesian". This is a general training-free strategy, no parameters are updated based on new observations. To me this gating mechanism is at most "adaptive".
+ **Sensitive hyperparameters setting.**
  + This method introduced many hyperparameters, and they are all adjustable: $L$, $\alpha\_{\text{max}}$, $k$, etc.
  + The hyperparameters are all set differently for different models and different benchmarks, showcasing the sensitiveness of them.
  + It's not clear how the author found the optimal setting of the hyperparameters. Is it based on an extra validation dataset?

**Questions:**

See above (Weaknesses).

---

> ### Author Response · Authors · 2025-11-21
>
> Thank you for the detailed feedback. We address each concern below.
>
> ### 1) What is the meaning of the main body of the steering vector $v_{\text{CARD}}$?
>
> $v_{\text{CARD}}$ is a **per-sample visual evidence direction**.
> During the mandatory prefill pass, we take the self-attention outputs at layer $\ell$,
> $$
> \Delta_i^{\ell} = A_i^{\ell},
> $$
> and aggregate them across all prefill tokens (image + prompt) to obtain a single direction:
>
> $$
> v_{\text{CARD}} = \mathrm{norm}\Big(\mathrm{Pool}_{\text{attn}}(A_i^{\ell})\Big).
> $$
>
> Intuitively, $v_{\text{CARD}}$ summarizes **how visual evidence flows into the decoder for this specific input**, and serves as the direction along which we apply residual steering.
>
> ---
>
> ### 2) Why is pooling $\Delta$ reasonable and not too lossy?
>
> We pool **attention outputs** to get a *stable global evidence direction*, not to preserve token-level information.
> Pooling is attention-aware, so visually grounded tokens receive higher weight, while low-content / syntactic tokens are down-weighted. This reduces noise and yields a robust evidence vector. Empirically, our geometry analyses show that this pooled direction tracks grounded generations and provides effective steering, indicating that the required information for steering is retained at the directional level.
>
> ---
>
> ### 3) Why steer *more* when $\cos(h, v_{\text{CARD}})$ is high?
>
> This is a key point: $g_t$ is a **trust / groundedness gate**, not a “deviation magnitude” detector.
>
> We always steer along a **fixed evidence direction** $v_{\text{CARD}}$:
> $$
> s_t = \cos(h_{\ell,t}, v_{\text{CARD}}), \qquad v_{\text{steer}} = g_t \cdot v_{\text{CARD}}.
> $$
> - When $s_t$ is high, the trajectory is already consistent with visual evidence, so **reinforcing along the same direction is safe and beneficial** (it strengthens grounded content without fighting the model’s local dynamics).
> - When $s_t$ is low/negative, this often occurs on non-visual tokens or local misalignment; strong updates there can **hurt fluency or recall**.
>
> Therefore, the gate intentionally **avoids over-steering where the evidence direction is locally untrustworthy**, while keeping a mild correction via bounded $[g_{\min}, g_{\max}]$.
>
> We will add one sentence in revision to clarify that $g_t$ measures *reliability of steering*, not “how wrong the model is.”
>
> ---
>
> ### 4) “Bayesian” terminology seems over-claimed
>
> We agree that our gating is not Bayesian in the sense of online weight updates.
> We use “Bayesian-inspired” to refer to the **structural form** of the gate derived from Beta–Bernoulli conjugacy: we model a latent groundedness variable $Z_t$ and take the posterior mean as the gate, with similarity providing pseudo-counts via softplus. This yields a monotone, symmetric, numerically stable mapping from $s_t$ to $g_t$ (App. A.2), rather than an ad-hoc heuristic.
>
> To avoid confusion, we will consistently rephrase this as a **“Bayesian-inspired adaptive gate.”**
>
> ---
>
> ### 5) Hyperparameter sensitivity and how we select them
>
> RUDDER has three **performance-critical** knobs: injection layer $L$, max strength $\alpha_{\max}$, and gate sharpness $k$.
> We select them on a **small holdout dev set (100 MSCOCO images)** to balance CHAIR reduction and recall, and **never tune on test sets**.
>
> The remaining parameters $g_{\min}, g_{\max}$, and $c$ are **engineering-oriented safety/behavior knobs**:
> - $g_{\min}, g_{\max}$ bound the update to prevent over-steering;
> - $c$ controls smoothness / prior concentration.
>
> We provide robust defaults, but they are intentionally left user-adjustable to trade off conservativeness vs. aggressiveness in deployment.
>
> Different backbones prefer different $L$ because their vision–language fusion stages differ; our per-model dev sweep over a narrow early-vs-late window is sufficient to identify $L$ with negligible cost.
>
> We will revise Sec. 4.1 to make this low-cost calibration protocol more explicit.

---

> > ### Comment · Reviewer_5Rux · 2025-11-23
> >
> > Thank you for your response. The authors' rebuttal has addressed most of my concerns. Please make sure the revision include:
> > + clearer rationale behind the gating mechanism;
> > + strategy for searching the hyperparameters and make sure the comparison is fair to the rest of the baselines.
> >
> > I've raised my score to 6.

---

### Official Review · Reviewer_56bn · 2025-11-02

**Soundness:** 2
**Presentation:** 3
**Contribution:** 2
**Rating:** 4
**Confidence:** 4

**Summary:**

This paper proposes RUDDER (Residual-Update Directed DEcoding Regulation), a method to mitigate object hallucination in Large Vision-Language Models (LVLMs). The method introduces two key components: (1) the CARD vector, a per-sample visual steering signal extracted at negligible computational cost during the prefill stage, and (2) the Beta Gate, an adaptive token-wise mechanism that dynamically adjusts intervention strength. While the experimental results appear promising, several aspects require clarification and further validation.

**Strengths:**

1. The paper clearly identifies the trade-off in existing methods—existing approaches incur high computational overhead and require multiple forward passes, which limits their practical deployment.

2.  The concept of dynamically adjusting intervention strength based on the model's deviation from visual context is well-motivated.

3. The paper is generally well-structured and clearly written, making it accessible to readers.

**Weaknesses:**

1. The Beta Gate design appears fundamentally counter-intuitive. According to Equation (3):

- When hl,t has high similarity with vCARD (i.e., cos⁡(hl,t, vCARD)≈1), the intervention strength g_t becomes large.
- When hl,t derivates from vCARD (i.e., cos⁡(hl,t, vCARD) is negative), the intervention strength g_t becomes small.
- This design contradicts common intuition: one would expect stronger intervention when the model deviates from visual grounding, not weaker. The paper does not adequately justify this seemingly backward design choice.


2. While the paper claims Beta Gate is "Bayesian-inspired," it lacks rigorous Bayesian derivation. The connection between the Beta distribution framework and the specific formulation in Equation (3) is unclear.

3. The paper omits several state-of-the-art hallucination mitigation methods. Missing references and performance comparisons (including effectiveness and efficiency): OPERA (Huang et al., 2023), HALC (Chen et al., 2024), ADHH (Yang et al., 2025).

OPERA: Alleviating Hallucination in Multi-Modal Large Language Models via Over-Trust Penalty and Retrospection-Allocation. CVPR 2023

HALC: Object Hallucination Reduction via Adaptive Focal-Contrast Decoding. ICML 2024.

Understanding and Mitigating Hallucinations in Large Vision-Language Models via Modular Attribution and Intervention. ICLR 2025.

**Questions:**

1. How are k (sensitivity) and c (concentration) determined? Why they are necessary?

2. Do hyperparameters need adjustment for different models (e.g., 7B vs. 13B)?

---

> ### Author Response · Authors · 2025-11-21
>
> Thank you for the detailed feedback. We address your concerns point-by-point below.
>
> ### Intuition of the steering vector and CARD
>
> **Meaning of $v_{\mathrm{CARD}}$.**
> CARD is a **per-sample visual evidence direction** extracted from the model’s **self-attention outputs** at a chosen decoder layer during the mandatory prefill pass. In a pre-norm decoder, the self-attention output equals the residual update:
> $$
> \Delta_i^{\ell} = A_i^{\ell}.
> $$
> We pool $\{A_i^{\ell}\}_{i \in T_{\mathrm{pre}}}$ over all prefill tokens (image + prompt) **via attention-aware pooling** (attention-weighted aggregation across tokens), then L2-normalize:
> $$
> v_{\mathrm{CARD}} = \mathrm{norm}\left( \mathrm{Pool}_{\mathrm{attn}}\left(A_i^{\ell}\right) \right).
> $$
> This gives a stable, input-specific direction summarizing the **net visual influence** for steering.
>
> Pooling is not intended to preserve token-level detail. It aggregates where the model places visual attention during prefill into a single robust evidence direction, while down-weighting low-content / syntactic tokens. Our geometry analyses verify that this direction aligns with effective grounding.
>
> ---
>
> ### “Counter-intuitive” gating direction
>
> We agree this can look non-standard if $g_t$ is read as a “deviation magnitude.” In RUDDER, $g_t$ is a **trust/groundedness gate on the reliability of the evidence direction**, not an error detector.
>
> We always steer **along the fixed evidence direction $v_{\mathrm{CARD}}$**. Let
> $$
> s_t = \cos\!\left(h_{\ell,t},\, v_{\mathrm{CARD}}\right).
> $$
> - If $s_t$ is high, the trajectory is already consistent with visual evidence, so **reinforcing along the same direction is safe and beneficial**.
> - If $s_t$ is low/negative, this often occurs on non-visual tokens or local misalignment; strong corrections there can **harm fluency/recall**.
>
> Hence the gate suppresses over-steering on untrustworthy steps, while clamping $[g_{\min}, g_{\max}]$ keeps a small but nonzero correction. We will add a short clarifying sentence in revision.
>
> ---
>
> ### “Bayesian-inspired” claim
>
> Thank you for the terminology check. We agree our method is not “Bayesian” in the sense of updating model weights. We use the term to describe the **structural derivation** of the gate from the **Beta–Bernoulli conjugate prior**.
>
> Instead of an ad-hoc mapping from cosine similarity to $[0,1]$, we view gating as estimating a latent “visual groundedness” probability $Z_t$:
> - alignment $s_t$ provides pseudo-counts $(\alpha_t, \beta_t)$;
> - $g_t$ is the posterior mean;
> - $k$ (sensitivity) and $c$ (concentration/smoothness) directly control this posterior, yielding a monotone, symmetric, numerically stable gate (App. A.2).
>
> We will revise the paper to consistently use **“Bayesian-inspired adaptive gate.”**
>
> ---
>
> ### Hyperparameters
>
> RUDDER has three performance-critical knobs: injection layer $L$, max strength $\alpha_{\max}$, and gate sharpness $k$. These are tuned on a **tiny holdout dev set (100 MSCOCO images)** to balance CHAIR and recall; we **never tune on test sets**.
>
> The remaining parameters are **engineering safety/behavior knobs**:
> - $g_{\min}, g_{\max}$: bound intervention to avoid over-steering;
> - $c$: gate smoothness / prior concentration.
>
> We provide robust defaults, and users may adjust them for stricter or stronger steering.
>
> This setup matches standard inference-time intervention practice: choosing an intervention site ($L$) and magnitude ($\alpha_{\max}$) is unavoidable, and we add only one extra adaptive knob ($k$). The calibration overhead is negligible because it is a single-pass method and the sweep is done on only 100 samples, once per backbone. We will make this low-cost protocol more prominent in Sec. 4.1.
>
> ---
>
> ### Other baselines and scalability
> You can find the additional experiments in the overall comment above. We didn't include OPERA (Huang et al., 2023) as a baseline because it has been surpassed by VISTA in their paper. We will add all the methods you mentioned as references in our revision. Our method is effective on larger models as well (e.g., 13B), but  ($\alpha_{\max}$) needs adjustment.

---

### Author Response · Authors · 2025-11-21

Thank you for handling our submission and for the thoughtful reviews. We appreciate the reviewers’ constructive feedback and have clarified the paper and added new experiments accordingly. Below, we summarize the key responses and updates.

**Core contribution & novelty.**
RUDDER is a **single-pass, training-free residual-stream steering method** for LVLM hallucination mitigation. It (i) extracts a **per-sample visual evidence direction** $v_{\text{CARD}}$ from self-attention residuals during the mandatory prefill pass, and (ii) performs token-wise adaptive injection via a **Beta Gate**. This targets a strong **efficacy–efficiency Pareto point** without extra forward passes.

**Key concerns addressed.**

1) **Method intuition & “Bayesian-inspired” terminology.**
We expanded the rationale for CARD pooling and clarified that the gate is **Bayesian-inspired in form (Beta–Bernoulli posterior-mean mapping)** rather than Bayesian weight-updating. We will consistently rename it a “Bayesian-inspired adaptive gate,” and clarify that $g_t$ measures **trust in steering** along $v_{\text{CARD}}$, not deviation magnitude.

2) **Hyperparameters & stability.**
RUDDER has three performance-critical knobs: injection layer $L$, max strength $\alpha_{\max}$, and gate sharpness $k$. They are tuned **once per backbone** on a **tiny 100-image dev set** under a CHAIR–recall constraint; we never tune on test sets. Other terms ($g_{\min}$, $g_{\max}$, $c$) are **engineering safety rails** with robust defaults. Ablations show broad stable regions rather than brittle spikes.

3) **Backbone-dependent layer choice.**
We clarified that optimal layers follow backbone fusion dynamics (late for early-fusion models with visual drift like LLaVA/Idefics2; early for strong early coupling or Q-Former distillation like Qwen2.5-VL/InstructBLIP), and added an architecture-aware selection guideline.

4) **Baselines and modern backbones (new experiments).**
Per reviewer request, we added comparisons with recent training-free baselines and evaluated on contemporary LVLMs. The new results are:

**LLaVA-1.5-7B**

|              | CHAIRs | CHAIRi |
| :----------- | ------ | ------ |
| Vanilla      | 48.6   | 13.6   |
| VISTA        | 38.6   | 11.4   |
| AGLA         | 44.6   | 12.1   |
| DeGF         | 34.2   | 9.24   |
| HALC         | 40.3   | 10.3   |
| RUDDER (Ours)| 39.5   | 10.5   |

Although DeGF obtains the lowest CHAIR, its overall latency is **>3×** the vanilla model. In contrast, RUDDER increases latency by **<4%**, supporting our main claim: strong hallucination reduction at near-vanilla speed.

**Qwen2.5-VL-7B**

|              | CHAIRs | CHAIRi |
| ------------ | ------ | ------ |
| Vanilla      | 35.2   | 9.5    |
| VISTA        | 25.1   | 7.7    |
| RUDDER (Ours)| 26.9   | 7.0    |

This confirms RUDDER remains effective on a modern open-source backbone.

**LLaVA-1.5-13B**

|              | CHAIRs | CHAIRi |
| ------------ | ------ | ------ |
| Vanilla      | 44.2   | 11.8   |
| RUDDER (Ours)| 39.9   | 10.8   |

This shows consistent gains as model scale increases.

5) **Faithfulness / diagnostic attribution.**
We already enforce **≥95% recall** to prevent trivial noun suppression and provide geometry evidence that CARD shifts representations away from text-only priors toward visual evidence. We will add a lightweight faithfulness-style diagnostic in revision to further substantiate causality.

**Takeaway.**
Across diverse backbones and decoding settings, RUDDER provides **substantial hallucination reduction under recall constraints** with **negligible overhead**, offering a practical inference-time solution. We hope these clarifications and new results resolve the reviewers’ concerns, and we thank you for your consideration.

---

### Meta-Review · Area_Chair_bPEw · 2026-01-06

**Summary:**

This paper proposes RUDDER, a training-free, inference-time steering method for mitigating object hallucinations in large vision–language models (LVLMs). The core idea is to extract a per-sample Contextual Activation Residual Direction (CARD) from self-attention residual updates during the mandatory prefill pass, and to inject this signal during decoding using a token-wise Bayesian-inspired adaptive gate. The method aims to strike a favorable efficiency–effectiveness trade-off, reducing hallucinations without incurring extra forward passes. Experiments on CHAIR and POPE across several LVLM backbones show reductions in hallucination metrics with negligible latency overhead.

Reviewers agreed that hallucination mitigation is an important problem and that RUDDER is well engineered and efficient. Several reviewers found the empirical results promising and appreciated the extensive ablations and added experiments in the rebuttal. However, despite these strengths, substantial concerns remain regarding the conceptual soundness of the gating mechanism, sensitivity to hyperparameters, modest gains on stronger backbones, and limited diagnostic evidence of faithfulness. These unresolved issues affect the strength and generality of the paper’s claims and motivate the rejection decision.

**Reviewer Concerns:**

Concerns that have been addressed satisfactorily:
- In response to concerns about missing baselines and backbone coverage raised by Reviewers 1UYi and 56bn: the authors added comparisons with recent training-free methods (e.g., DeGF, AGLA, HALC) and evaluated on additional backbones, including Qwen2.5-VL and larger LLaVA variants.
- In response to concerns about clarity of CARD construction and pooling raised by Reviewers 56bn and XUvK: the authors clarified which residual tensors are pooled, how attention-weighted pooling is performed, and provided additional geometric analyses supporting the stability of the pooled direction.
- In response to concerns about the “Bayesian” terminology raised by multiple reviewers: the authors acknowledged that the method is not Bayesian in a strict sense and committed to consistently using the term “Bayesian-inspired adaptive gate,” clarifying the intended interpretation.
- In response to concerns about efficiency and practicality raised by Reviewers 5Rux and XUvK: the authors provided detailed latency and throughput analyses showing that RUDDER introduces negligible overhead compared to vanilla decoding.

Concerns that have not been addressed satisfactorily:
- Conceptual soundness of the gating mechanism (Reviewer 56bn, Reviewer 5Rux): reviewers questioned the seemingly counter-intuitive design choice of applying stronger steering when hidden states are already well aligned with visual evidence, rather than when they deviate. While the rebuttal reframed the gate as a “trust” mechanism, this explanation remains heuristic and does not fully resolve concerns about whether the design is principled or generally applicable.
- Hyperparameter sensitivity and deployment burden (Reviewer 5Rux, Reviewer XUvK, Reviewer 1UYi): although the authors clarified a calibration protocol, the method still relies on multiple model-specific hyperparameters (e.g., injection layer, gate sharpness, strength bounds), with different optimal settings across backbones. This raises concerns about robustness and ease of deployment in practice.
- Modest gains on stronger LVLMs (Reviewer 1UYi): the added experiments indicate that performance improvements diminish as the underlying LVLM becomes stronger, suggesting limited marginal benefit on modern, well-grounded models.
- Lack of strong faithfulness diagnostics (Reviewer XUvK): while outcome metrics and geometric analyses were provided, reviewers remained unconvinced that the method truly reduces reliance on language priors rather than suppressing certain token patterns. The promised faithfulness diagnostics remain preliminary.

**Reviewer Scores:**

- Reviewer 56bn: Marginally below threshold (4); would likely maintain the score given unresolved concerns about conceptual soundness.
- Reviewer 5Rux: Increased score from 4 to 6 after rebuttal; would likely maintain this score, leaning positive but not strongly advocating acceptance.
- Reviewer XUvK: Maintained a marginally positive score (6); would likely keep the score, acknowledging strengths but remaining cautious.
- Reviewer 1UYi: Negative (2); would maintain the score, as concerns about diminishing gains and dated baselines persist.

---

### Decision · Program_Chairs · 2026-01-26

Reject